# Mapping sequence structure in the human lateral entorhinal cortex

Jacob LS Bellmund[1,2,3]*, Lorena Deuker[4], Christian F Doeller[1,2]*

[1]Max Planck Institute for Human Cognitive and Brain Sciences, Leipzig, Germany; [2]Kavli Institute for Systems Neuroscience, Norwegian University of Science and Technology, Trondheim, Norway; [3]Donders Institute for Brain, Cognition and Behaviour, Radboud University, Nijmegen, Netherlands; [4]Department of Neuropsychology, Institute of Cognitive Neuroscience, Faculty of Psychology, Ruhr University Bochum, Bochum, Germany

**Abstract** Remembering event sequences is central to episodic memory and presumably supported by the hippocampal-entorhinal region. We previously demonstrated that the hippocampus maps spatial and temporal distances between events encountered along a route through a virtual city (Deuker et al., 2016), but the content of entorhinal mnemonic representations remains unclear. Here, we demonstrate that multi-voxel representations in the anterior-lateral entorhinal cortex (alEC) — the human homologue of the rodent lateral entorhinal cortex — specifically reflect the temporal event structure after learning. Holistic representations of the sequence structure related to memory recall and the timeline of events could be reconstructed from entorhinal multi-voxel patterns. Our findings demonstrate representations of temporal structure in the alEC; dovetailing with temporal information carried by population signals in the lateral entorhinal cortex of navigating rodents and alEC activations during temporal memory retrieval. Our results provide novel evidence for the role of the alEC in representing time for episodic memory.

DOI: https://doi.org/10.7554/eLife.45333.001

*For correspondence:
bellmund@cbs.mpg.de (JLSB);
doeller@cbs.mpg.de (CFD)

**Competing interests:** The authors declare that no competing interests exist.

## Introduction

Knowledge of the temporal structure of events is central to our experience. We remember how a sequence of events unfolded and can recall when in time events occurred. Emphasizing both when in time and where in space events came to pass, episodic memories typically comprise event information linked to a spatiotemporal context. Space and time have been suggested to constitute fundamental dimensions along which our experience is organized (*Konkel and Cohen, 2009*; *Ekstrom and Ranganath, 2018*; *Bellmund et al., 2018a*). Consistently, the role of the hippocampus — a core structure for episodic memory (*Scoville and Milner, 1957*; *Squire, 1982*) — in coding locations in space (*O'Keefe and Dostrovsky, 1971*; *Moser et al., 2017*; *Epstein et al., 2017*) and moments in time (*Pastalkova et al., 2008*; *MacDonald et al., 2011*; *Eichenbaum, 2014*; *Ranganath, 2019*; *Howard, 2018*) is well-established. Human memory research has highlighted the role of the hippocampus in the encoding, representation and retrieval of temporal relations (*Tubridy and Davachi, 2011*; *DuBrow and Davachi, 2014*; *Ezzyat and Davachi, 2014*; *Hsieh et al., 2014*; *Jenkins and Ranganath, 2010*; *Jenkins and Ranganath, 2016*; *Kyle et al., 2015*; *Copara et al., 2014*). The similarity patterns of mnemonic representations suggest that the hippocampus forms integrated maps reflecting the temporal and spatial structure of event memories (*Deuker et al., 2016*; *Nielson et al., 2015*). Consistently, activity in the hippocampal-entorhinal region has been demonstrated to be sensitive to Euclidean distances as well as the lengths of shortest paths to goals

during navigation (*Spiers and Maguire, 2007*; *Viard et al., 2011*; *Sherrill et al., 2013*; *Howard et al., 2014*; *Chrastil et al., 2015*; *Spiers and Barry, 2015*).

How do representations of temporal structure arise in the hippocampus? Evidence suggests that neural ensembles in the lateral entorhinal cortex (EC), which is strongly connected to the hippocampus (*Witter et al., 2017*), carry temporal information in freely moving rodents (*Tsao et al., 2018*). Specifically, temporal information could be decoded from population activity with high accuracy (*Tsao et al., 2018*). This temporal information was suggested to arise from the integration of experience rather than an explicit clocking signal (*Tsao et al., 2018*). Recently, the human anterior-lateral entorhinal cortex (alEC), the homologue region of the rodent lateral entorhinal cortex (*Navarro Schröder et al., 2015*; *Maass et al., 2015*), as well as the perirhinal cortex and a network of brain regions including the hippocampus, the medial prefrontal cortex, posterior cingulate cortex and angular gyrus have been implicated in the recall of temporal information (*Montchal et al., 2019*). These regions responded more strongly for high compared to low accuracy retrieval of when in time snapshots from a sitcom appeared over the course of the episode viewed in the experiment (*Montchal et al., 2019*). Together, these findings demonstrate that entorhinal population activity carries temporal information in navigating rodents and that its human homologue is activated during temporal memory recall. However, the contents of mnemonic representations in the alEC remains unclear.

We used representational similarity analysis of fMRI multi-voxel patterns in the entorhinal cortex to address the question how learning the structure of an event sequence shapes mnemonic representations in the alEC. Using this paradigm and data, we previously demonstrated that participants can successfully recall spatial and temporal relations of events defined by object encounters in a virtual city and that the change of hippocampal representations reflects an integrated event map of the remembered distance structure (*Deuker et al., 2016*). Here, we show that the change of multi-voxel pattern similarity through learning in the alEC specifically reflects the temporal structure of the event sequence.

## Results

We examined the effect of learning on object representations in the human entorhinal cortex using fMRI. In between two picture viewing tasks during which fMRI data were collected, participants acquired knowledge of temporal and spatial positions of objects in a familiar virtual city. Participants navigated repeated laps of a route along which they encountered chests containing different objects (*Figure 1*; *Figure 1—figure supplement 1*). We aimed to test whether entorhinal pattern similarity change from before to after learning related to experienced object relationships. Specifically, we presented object images twelve times in the picture viewing tasks before and after learning, using the same random order in both scanning runs. For both runs, we calculated the similarity of multi-voxel patterns for all object pairs and correlated changes in representational similarity with the temporal and spatial object relationships. The temporal distance structure of the object sequence can be quantified as the elapsed time between object encounters or as ordinal differences between their sequence positions, which are closely related in our task. Spatial distances on the other hand can be captured by Euclidean distances or geodesic distances between positions based on the shortest navigable paths between object positions. Importantly, we dissociated temporal from Euclidean and geodesic spatial object relationships through the use of teleporters along the route (*Figure 1—figure supplement 2*). Further, object relationships can be quantified by the distance traveled along the section of the route separating their positions (*Figure 1—figure supplement 2*). To assess whether entorhinal object representations change from before to after learning to map experienced object relationships, we compared changes in neural pattern similarity to the temporal and spatial structure of the task.

The change in multi-voxel pattern similarity in alEC between pre- and post-learning scans was negatively correlated with the sequence structure (*Figure 2A* and *Figure 2B*, T(25)=- 3.75, p=0.001, alpha-level of 0.0125, Bonferroni-corrected for four comparisons), which was quantified as the median elapsed time between objects pairs along the route. After relative to before learning, objects encountered in temporal proximity were represented more similarly compared to object

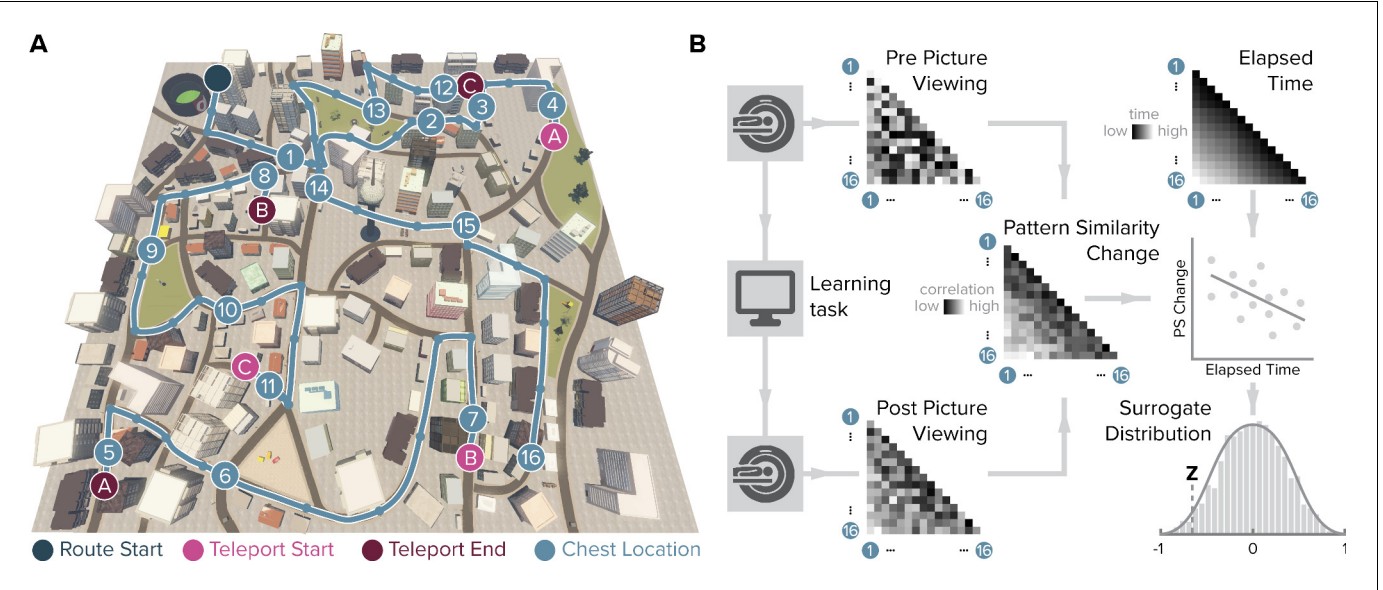

**Figure 1.** Design and analysis logic. (**A**) During the spatio-temporal learning task, which took place in between two identical runs of a picture viewing task (*Figure 1—figure supplement 1*), participants repeatedly navigated a fixed route (blue line, mean ± standard deviation of median time per lap 264.6 ± 47.8 s) through the virtual city along which they encountered objects hidden in chests (numbered circles) (*Deuker et al., 2016*). Temporal (median time elapsed) and spatial (Euclidean and geodesic) distances between objects were dissociated through the use of three teleporters (lettered circles) along the route (*Figure 1—figure supplement 2*), which instantaneously changed the participant's location to a different part of the city. (**B**) In the picture viewing tasks, participants viewed randomly ordered images of the objects encountered along the route while fMRI data were acquired. We quantified multi-voxel pattern similarity change between pairwise object comparisons from before to after learning the temporal and spatial relationships between objects in subregions of the entorhinal cortex. We tested whether pattern similarity change reflected the structure of the event sequence, by correlating it with the time elapsed between objects pairs (top right matrix shows median elapsed time between object encounters along the route averaged across participants). For each participant, we compared the correlation between pattern similarity change and the prediction matrix to a surrogate distribution obtained via bootstrapping and used the resulting z-statistic for group-level analysis (see Materials and methods).
DOI: https://doi.org/10.7554/eLife.45333.002

The following figure supplements are available for figure 1:

**Figure supplement 1.** Overview of experimental design.
DOI: https://doi.org/10.7554/eLife.45333.003
**Figure supplement 2.** Temporal distances are not correlated with Euclidean or geodesic spatial distances.
DOI: https://doi.org/10.7554/eLife.45333.004

pairs further separated in time (*Figure 2C*). Pattern similarity change was negatively correlated with temporal distances after excluding comparisons of objects encountered in direct succession from the analysis (T(25)=-2.00, p=0.029, one-sided test, *Figure 2—figure supplement 1A*), speaking for holistic representations of temporal structure in the alEC and ruling out that the effect we observed is largely driven by increased similarity of temporally adjacent objects. Importantly, the strength of this effect was strongly related to behavior in the post-scan free recall test, where participants retrieved the objects from memory. Specifically, participants with stronger correlations between alEC pattern similarity change and the temporal task structure tended to recall objects together that were encountered in temporal proximity along the route (Pearson r = −0.53, p=0.006, CIs: −0.76,–0.19, *Figure 2D*).

Pattern similarity change in alEC did not correlate significantly with Euclidean spatial distances (T(25)=0.81, p=0.420) and pattern similarity change in posterior-medial EC (pmEC) did not correlate with Euclidean (T(25)=0.58, p=0.583) or temporal (T(25)=1.73, p=0.089) distances. Temporal distances between objects during the first picture viewing task were not related to alEC pattern similarity change (*Figure 2—figure supplement 1B*; T(24)=-0.29 p=0.776, one outlier excluded, see Materials and methods) and correlations with elapsed time between objects during navigation were significantly more negative (T(24)=-1.76 p=0.045; one-sided test); strengthening our interpretation that pattern similarity changes reflected relationships experienced in the virtual city.

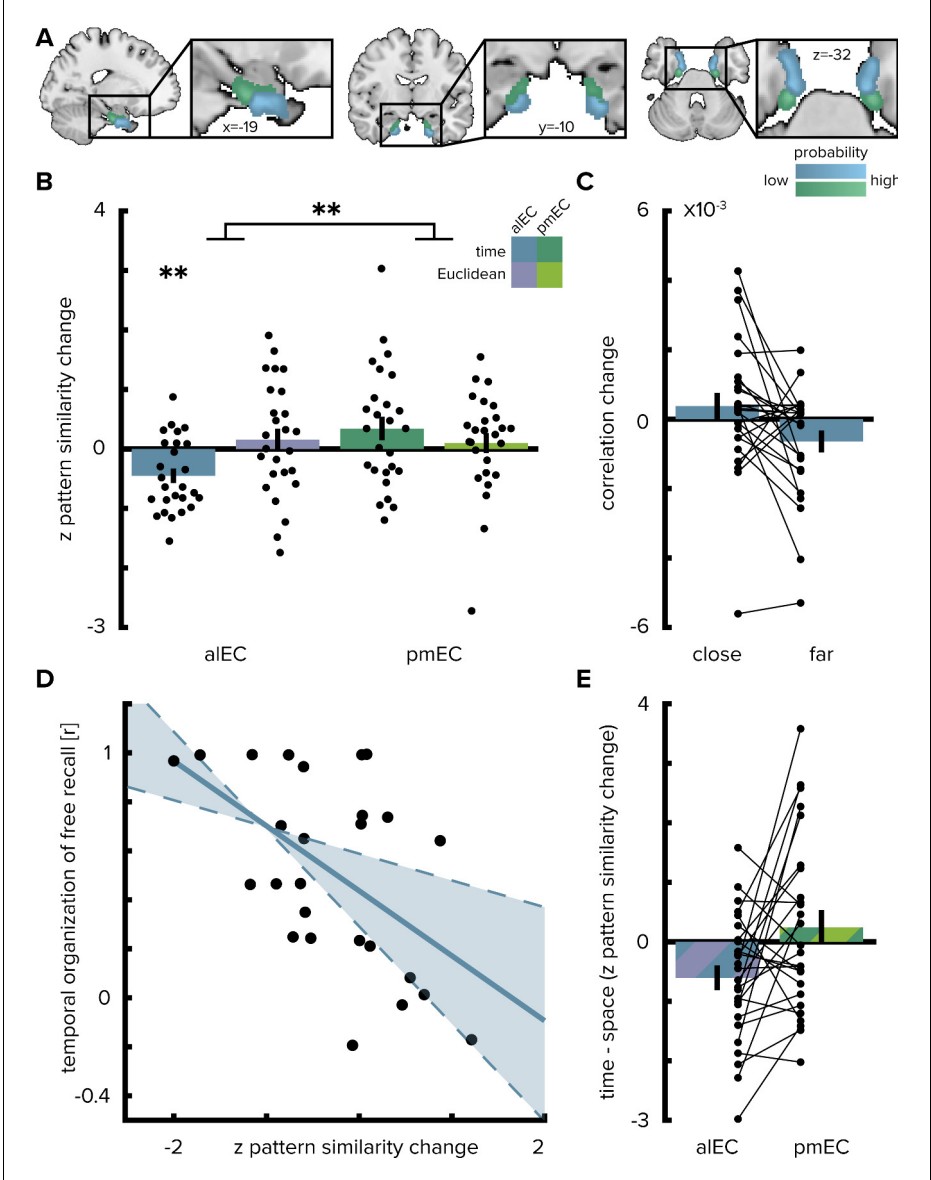

Figure 2. Temporal mapping in alEC. (A) Entorhinal cortex subregion masks from *Navarro Schröder et al. (2015)* were moved into subject-space and intersected with participant-specific Freesurfer parcellations of entorhinal cortex. Color indicates probability of voxels to belong to the alEC (blue) or pmEC (green) subregion mask after subject-specific masks were transformed back to MNI template space for visualization. (B) Pattern similarity change in the alEC correlated with the temporal structure of object relationships, defined by the median time elapsed between object encounters, as indicated by z-statistics significantly below 0. A permutation-based two-way repeated measures ANOVA further revealed a significant interaction highlighting a difference in mapping temporal and Euclidean spatial distances between alEC and pmEC. (C) To break down the negative correlation of alEC pattern similarity change and temporal distance shown in (B), pattern similarity change is plotted separately for object pairs close together or far apart in time along the route based on a median split of elapsed time between object encounters. (D) Pattern similarity change in alEC was negatively related to temporal relationships independent of objects encountered in direct succession (*Figure 2—figure supplement 1A*). The magnitude of this effect correlated significantly with participants' free recall behavior. The temporal organization of freely recalled objects was assessed by calculating the absolute difference in position for all recalled objects and correlating this difference with the time elapsed between encounters of object pairs along the route. Solid line shows least squares line; dashed lines and shaded region highlight bootstrapped confidence intervals. (E) To illustrate the interaction effect shown in (B), the difference in the relationship between temporal and spatial distances to pattern similarity change is shown for alEC and pmEC. Negative values indicate stronger correlations

*Figure 2 continued on next page*

*Figure 2 continued*

with temporal compared to spatial distances. Bars show mean and S.E.M with lines connecting data points from the same participant in (**C** and **E**). **p<0.01.

DOI: https://doi.org/10.7554/eLife.45333.005

The following source data and figure supplements are available for figure 2:

**Source data 1.** Z-values of correlations between pattern similarity change in the entorhinal subregions and temporal and Euclidean spatial distances as shown in panel B.

DOI: https://doi.org/10.7554/eLife.45333.010

**Source data 2.** Pattern similarity changes in alEC for object pairs separated by low and high temporal distances as shown in panel C.

DOI: https://doi.org/10.7554/eLife.45333.011

**Source data 3.** Z-values of correlations between alEC pattern similarity change and temporal distances without comparisons of objects encountered in direct succession along the route and Pearson correlation coefficients quantifying temporal clustering during the free recall task (panel D).

DOI: https://doi.org/10.7554/eLife.45333.012

**Source data 4.** Z-value differences quantifying the difference in temporal and spatial mapping in alEC and pmEC as shown in panel E.

DOI: https://doi.org/10.7554/eLife.45333.013

**Figure supplement 1.** Entorhinal pattern similarity change reflects temporal structure beyond direct adjacency and stimulus presentation times from the pre-learning scan.

DOI: https://doi.org/10.7554/eLife.45333.006

**Figure supplement 2.** Geodesic spatial distances do not correlate with entorhinal pattern similarity change.

DOI: https://doi.org/10.7554/eLife.45333.007

**Figure supplement 3.** Signal-to-noise ratio in the entorhinal cortex.

DOI: https://doi.org/10.7554/eLife.45333.008

**Figure supplement 4.** No evidence for reactivation of object representations in the entorhinal cortex.

DOI: https://doi.org/10.7554/eLife.45333.009

Can we reconstruct the timeline of events from pattern similarity change in alEC? Here, we used multidimensional scaling to extract coordinates along one dimension from pattern similarity change averaged across participants (*Figure 3A–D*). The reconstructed temporal coordinates, transformed into the original value range using Procrustes analysis (*Figure 3A*), mirrored the time points at which objects were encountered during the task (*Figure 3B*, Pearson correlation between reconstructed and true time points, r = 0.56, p=0.023, bootstrapped 95% confidence interval: 0.21, 0.79). Further, we contrasted the fit of the coordinates from multidimensional scaling between the true and randomly shuffled timelines (*Figure 3C*). Specifically, we compared the standardized sum of squared errors of the fit between the reconstructed and the true timeline, the Procrustes distance, to a surrogate distribution of deviance values. This surrogate distribution was obtained by fitting the coordinates from multidimensional scaling to randomly shuffled timelines of events. The Procrustes distance from fitting to the true timeline was smaller than the 5th percentile of the surrogate distribution generated via 10000 random shuffles (*Figure 3D*, p=0.026). Taken together, these findings indicate that alEC representations change through learning to reflect the temporal structure of the acquired event memories and that we can recover the timeline of events from this representational change.

What is the nature of regional specificity within entorhinal cortex? In a next step, we compared temporal and spatial mapping between the subregions of the entorhinal cortex. We conducted a permutation-based two-by-two repeated measures ANOVA (see Materials and methods) with the factors entorhinal subregion (alEC vs. pmEC) and relationship type (temporal vs. Euclidean spatial distance between events). Crucially, we observed a significant interaction between EC subregion and distance type ($F_{(1,25)}=7.40$, p=0.011, *Figure 2B and E*). Further, the main effect of EC subregion was significant ($F_{(1,25)}=5.18$, p=0.029), while the main effect of distance type was not ($F_{(1,25)}=0.84$, p=0.367). Based on the significant interaction, we conducted planned post-hoc comparisons, which revealed significant differences (Bonferroni-corrected alpha-level of 0.025) between the mapping of temporal and spatial distances in alEC ($T_{(25)}=-2.91$, p=0.007) and a significant difference between temporal mapping in alEC compared to pmEC ($T_{(25)}=-3.52$, p=0.001).

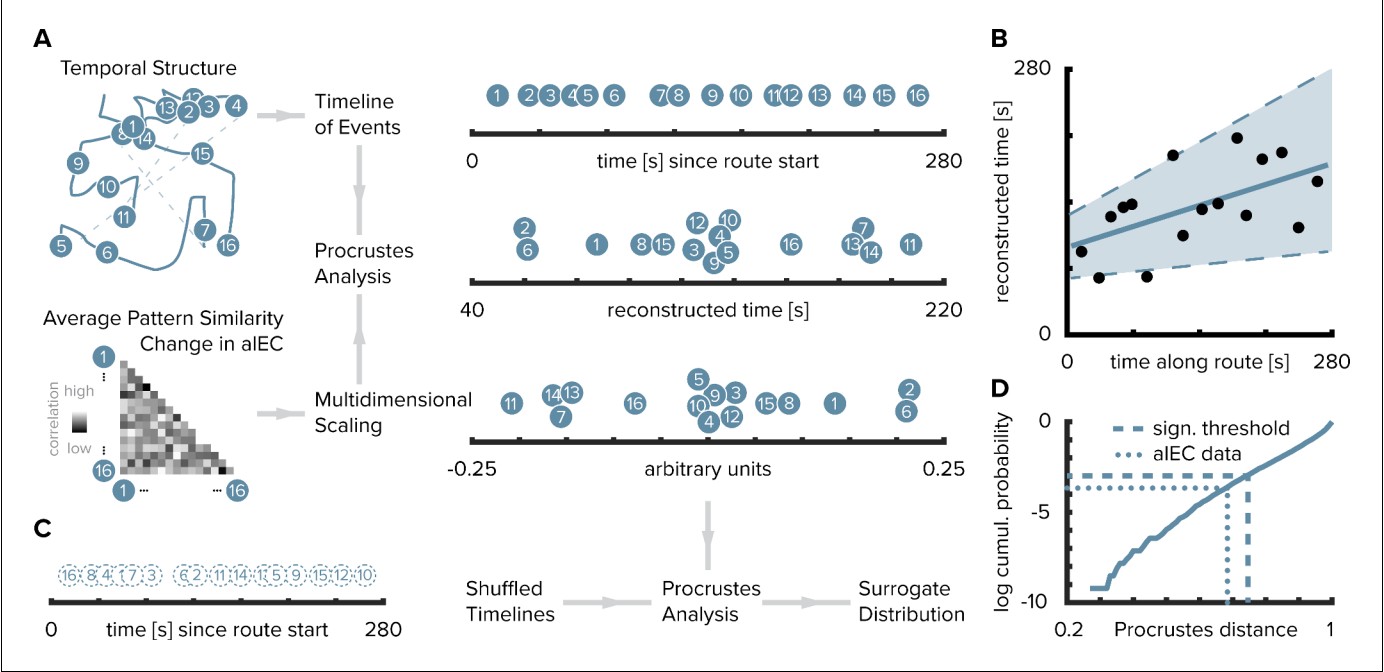

**Figure 3.** Reconstructing the timeline of events from entorhinal pattern similarity change. (**A**) To recover the temporal structure of events we performed multidimensional scaling on the average pattern similarity change matrix in alEC. The resulting coordinates, one for each object along the route, were subjected to Procrustes analysis, which applies translations, rotations and uniform scaling to superimpose the coordinates from multidimensional scaling on the true temporal coordinates along the route (see Materials and methods). For visualization, we varied the positions resulting from multidimensional scaling and Procrustes analysis along the y-axis. (**B**) The temporal coordinates of this reconstructed timeline were significantly correlated with the true temporal coordinates of object encounters along the route. Circles indicate time points of object encounters; solid line shows least squares line; dashed lines and shaded region highlight bootstrapped confidence intervals. (**C**) The goodness of fit of the reconstruction (the Procrustes distance) was quantified as the standardized sum of squared errors and compared to a surrogate distribution of Procrustes distances. This surrogate distribution was obtained from randomly shuffling the true coordinates against the coordinates obtained from multidimensional scaling and then performing Procrustes analysis for each of 10000 shuffles (left shows one randomly shuffled timeline for illustration). (**D**) The Procrustes distance obtained from fitting to the true timeline of events (dotted line) was smaller than the 5th percentile (dashed line) of the surrogate distribution (solid line), which constitutes the significance threshold at an alpha level of 0.05.

DOI: https://doi.org/10.7554/eLife.45333.014

The following source data is available for figure 3:

**Source data 1.** True and reconstructed temporal coordinates of object positions as shown in panel B.
DOI: https://doi.org/10.7554/eLife.45333.015

**Source data 2.** Procrustes distance from mapping coordinates from multidimensional scaling based on alEC pattern similarity change to true temporal coordinates and surrogate distribution obtained from fitting to shuffled temporal coordinates (panel D).
DOI: https://doi.org/10.7554/eLife.45333.016

Operationalizing the temporal structure in terms of the ordinal distances between object positions in the sequence yielded comparable results since our design did not disentangle time elapsed from ordinal positions as objects were encountered at regular intervals along the route. Pattern similarity change in alEC correlated significantly with ordinal temporal distances (**Figure 4**, T(25)=-3.37, p=0.002), an effect further qualified by a significant interaction in the two-by-two repeated measures ANOVA contrasting the effects of ordinal temporal and Euclidean spatial distances in the entorhinal subregions (interaction: F(1,25)=7.11, p=0.012; main effect of EC subregion: F(1,25)=4.97, p=0.033; main effect of distance type: F(1,25)=0.84, p=0.365). Alternative to the quantification of spatial relationships as Euclidean distances we calculated geodesic distances between object positions (**Figure 1—figure supplement 2B–E**). Entorhinal pattern similarity change was not correlated with geodesic distances based on shortest paths between locations using all positions not obstructed by buildings or other obstacles (**Figure 2—figure supplement 2A**, alEC: T(25)=0.82, p=0.436, pmEC: T(25)=0.73, p=0.479) or based on shortest paths using only the street network (**Figure 2—figure supplement 2B**, alEC: T(25)=0.36, p=0.715, pmEC: T(25)=0.92, p=0.375). Furthermore, the interaction

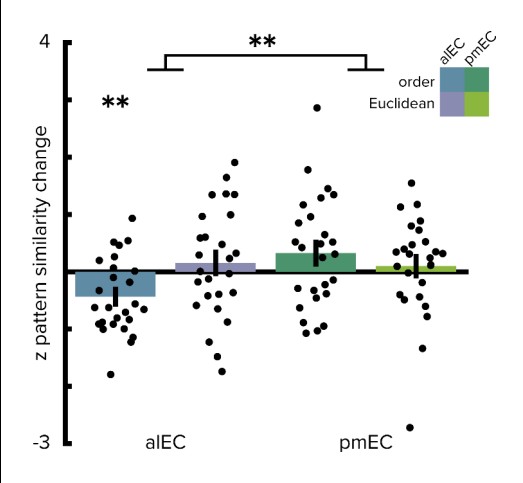

**Figure 4.** Ordinal temporal distances correlate with pattern similarity change in alEC. Repeating the two-way repeated measures ANOVA using ordinal distances as the measure of sequence structure yielded results comparable to the analyses presented in *Figure 2*. We observed a significant interaction (see main text) highlighting a difference in temporal and spatial mapping between alEC and pmEC. Post hoc tests comparing mapping of ordinal temporal distances and Euclidean spatial distances in the alEC (T(25)=-2.81, p=0.008) and comparing mapping of ordinal temporal distances between alEC and pmEC (T(25)=-3.53, p=0.002) are significant at the Bonferroni-corrected alpha-level of 0.025. Bars reflect mean and S.E.M with circles showing data points of individual participants. **p<0.01.

DOI: https://doi.org/10.7554/eLife.45333.017

The following source data is available for figure 4:

**Source data 1.** Z-values of correlations between pattern similarity change in the entorhinal subregions and ordinal temporal and Euclidean spatial distances.

DOI: https://doi.org/10.7554/eLife.45333.018

of the two-by-two repeated measures ANOVA with the factors entorhinal subregion and distance type remained significant when using geodesic spatial distances based on shortest paths using all non-obstructed positions (interaction: F(1,25)=6.96, p=0.014; main effect of EC subregion: F(1,25)=5.18, p=0.031; main effect of distance type: F(1,25)=0.99, p=0.330) or the street network only (interaction: F(1,25)=4.30, p=0.048; main effect of EC subregion: F(1,25)=6.68, p=0.017; main effect of distance type: F(1,25)=0.81, p=0.376). Spatial and temporal signal-to-noise ratios did not differ between alEC and pmEC (*Figure 2—figure supplement 3*), ruling out that differences in signal quality might explain the observed effects.

Does the presentation of object images during the post-learning picture viewing task elicit reactivations of similar representations from the pre-scan? For example, associations might be formed between objects encountered in succession along the route, which might result in the reactivation of neighboring objects after learning. To test this notion, we trained pattern classifiers to distinguish object representations on the pre-learning scan and tested these classifiers on the post-learning scan (see Materials and methods). We observed classifier accuracies exceeding chance levels in the lateral occipital cortex (LOC, *Figure 2—figure supplement 4*, T(25)=7.54, p<0.001, see Materials and methods) — known to be involved in visual object processing (*Grill-Spector et al., 2001*). In the entorhinal cortex, classification accuracies did not exceed chance levels (alEC: T(25)=-0.08, p=0.941; pmEC: T(25)=0.53, p=0.621). Next, we examined classifier predictions as a function of lag along the sequence. If effects in the alEC are driven by the reactivation of objects at neighboring sequence positions, then one might expect systematic classification errors, where an object might likely be confused with preceding or successive objects. In the entorhinal cortex, classifier evidence did not exceed chance levels for the three objects preceding (*Figure 2—figure supplement 4B*, alEC: most extreme T(25)=1.13; minimum p=0.270; pmEC: most extreme T(25)=1.00; min. p=0.332) or following (alEC: most extreme T(25)=-2.07; min. p=0.055; pmEC: most extreme T(25)=-0.83; min. p=0.414) an object. We also did not observe above-chance classifier evidence for nearby objects in the LOC, but

rather classifier evidence was below chance levels for some lags, potentially due to high classification accuracies at no lag (preceding objects: most extreme T(25)=-2.51; min. p=0.018; successive objects: most extreme T(25)=-4.09; min. p<0.001).

Collectively, our findings demonstrate that, within the EC, only representations in the anterior-lateral subregion changed to resemble the temporal structure of the event sequence and that this mapping was specific to the temporal rather than the spatial dimension.

## Discussion

We examined the similarity of multi-voxel patterns to demonstrate that alEC representations reflect the experienced temporal event structure. Despite being cued in random order after learning, these representations related to a holistic temporal map of the sequence structure. Moreover, entorhinal pattern similarity change correlated with participants' recall behavior and we recovered the timeline of events during learning from these representations.

Our hypothesis for temporal mapping in the alEC specifically was based on a recent finding demonstrating that population activity in the rodent lateral EC carries information from which time can be decoded at different scales ranging from seconds to days (*Tsao et al., 2018*). Time could be decoded with higher accuracies from the lateral EC than the medial EC and hippocampal subfield CA3. During a structured task in which the animal ran repeated laps on a maze separated into different trials, neural trajectories through population activity space were similar across trials, illustrating that the dynamics of lateral EC neural signals were more stable than during free foraging (*Tsao et al., 2018*). Consistently, temporal coding was improved for time within a trial during the structured task compared to episodes of free foraging. These findings support the notion that temporal information in the lateral EC might inherently arise from the encoding of experience (*Tsao et al., 2018*). In our task, relevant factors contributing to a similar experience of the route on each lap are not only the encounters of objects in a specific order at their respective positions, but also recognizing and passing salient landmarks as well as traveled distance and navigational demands in general. Changes in metabolic states and arousal presumably varied more linearly over time. Slowly drifting activity patterns have been observed also in the human medial temporal lobe (*Folkerts et al., 2018*) and EC specifically (*Lositsky et al., 2016*). A representation of time within a known trajectory in the alEC could underlie the encoding of temporal relationships between events in our task, where participants repeatedly navigated along the route to learn the positions of objects. Hence, temporal mapping in the alEC as we report here might help integrate hippocampal spatio-temporal event maps (*Deuker et al., 2016*).

Our findings demonstrate that alEC representations reflect the temporal structure of events after learning. This finding further dovetails with a recent fMRI study (*Montchal et al., 2019*), in which participants indicated when a still frame was encountered over the course of an episode of a sitcom. The alEC activated more strongly for the third of trials in which participants recalled temporal information most accurately compared to the third of trials in which temporal precision was lowest (*Montchal et al., 2019*). Going beyond the relationship of univariate activation differences to the precision of temporal memory recall, we focused on the content of alEC activation patterns and demonstrate that the alEC represents the temporal structure of events after learning.

One possibility for why alEC multi-voxel patterns resemble a holistic temporal map of the event structure in our task is the reactivation of temporal context information. If alEC neural populations traverse similar population state trajectories on each lap, they would carry information about time within a lap. A given object would be associated with a similar alEC population state on each lap. Associations with temporally drifting signals during the learning task would result in representational changes relative to the baseline scan that, if reactivated in the post-learning picture viewing task, reflect the experienced temporal structure of object encounters. This might explain the observed pattern similarity structure with relatively increased similarity for objects encountered in temporal proximity during learning and decreased similarity for items encountered after longer delays. While this interpretation is in line with data from rodent electrophysiology (*Tsao et al., 2018*) and the framework proposed by the temporal context model (*Howard and Kahana, 2002*; *Howard et al., 2005*) as well as evidence for neural contiguity effects in image recognition tasks (*Howard et al., 2012*; *Folkerts et al., 2018*), we cannot test the reinstatement of specific activity patterns from the

learning phase directly since fMRI data were only collected during the picture viewing tasks in this study.

An alternative explanation for how the observed effects might arise is through associations between the objects. During learning, an object might become associated with preceding and successive objects, with stronger associations for objects closer in the sequence (*Metcalfe and Murdock, 1981*; *Lewandowsky and Murdock, 1989*; *Jensen and Lisman, 2005*). In this framework, the reactivation of associated objects during the post-learning picture viewing task could drive similarity increases for objects close together in the sequence. We tested for stable object representations from before to after learning and assessed classifier predictions to test the hypothesis that — if object reactivations underlie the effects — we might observe biased classifier evidence for the objects preceding or following a given object in the sequence. However, using classifiers trained on the picture viewing task before learning, we did not observe evidence for stable object representations in the entorhinal cortex or above-chance classifier evidence for objects nearby in the sequence after learning. Object representations in lateral occipital cortex (LOC) were stable between the picture viewing tasks. Previous studies have observed evidence for cortical reinstatement during memory retrieval (*Nyberg et al., 2000*; *Wheeler et al., 2000*; *Polyn et al., 2005*), modulated by hippocampal-entorhinal activity (*Bosch et al., 2014*). We did not observe classification accuracies exceeding chance levels for objects from nearby sequence positions in LOC, which one would expect if associative retrieval of objects accompanied by cortical reinstatement were to underlie our effects. Hence, these results fail to provide evidence for the notion that the reactivation of object representations drove our effects.

Importantly, the highly-controlled design of our study supports the interpretation that alEC representations change through learning to map the temporal event structure. The order of object presentations during the scanning sessions was randomized and thus did not reflect the order in which objects were encountered during the learning task. Since the assignment of objects to positions was randomized across participants and we analyzed pattern similarity *change* from a baseline scan, our findings do not go back to prior associations between the objects, but reflect information learned over the course of the experiment. Further, we presented the object images during the scanning sessions not only in the same random order, but also with the same presentation times and inter-stimulus intervals; thereby ruling out that the effects we observed relate to temporal autocorrelation of the BOLD-signal. Taken together, the high degree of experimental control of our study supports the conclusion that alEC representations change to reflect the temporal structure of acquired memories.

The long time scales of lateral EC temporal codes differ from the observation of time cells in the hippocampus and medial EC, which fire during temporal delays in highly trained tasks (*Pastalkova et al., 2008*; *MacDonald et al., 2011*; *Eichenbaum, 2014*; *Kraus et al., 2015*; *Mau et al., 2018*; *Heys and Dombeck, 2018*). Time cell ensembles change over minutes and days (*Mau et al., 2018*), but their firing has been investigated predominantly in the context of short delays in the range of seconds. One recent study did not find evidence for time cell sequences during a 60s-delay (*Sabariego et al., 2019*). In our task, one lap of the route took approximately 4.5 min on average; comparable to the 250s-duration of a trial in *Tsao et al. (2018)*. How memories are represented at different temporal scales, which might be integrated in hierarchically nested sequences such as different days within a week, remains a question for future research.

Our assessment of temporal representations in the antero-lateral and posterior-medial subdivision of the EC was inspired by a recent report of temporal coding during free foraging and repetitive behavior in the rodent EC, which was most pronounced in the lateral EC (*Tsao et al., 2018*). In humans, local and global functional connectivity patterns suggest a preserved bipartite division of the EC, but along not only its medial-lateral, but also its anterior-posterior axis (*Navarro Schröder et al., 2015*; *Maass et al., 2015*). Via these entorhinal subdivisions, cortical inputs from the anterior-temporal and posterior-medial memory systems might converge onto the hippocampus (*Ranganath and Ritchey, 2012*; *Ritchey et al., 2015*). In line with hexadirectional signals in pmEC during imagination (*Bellmund et al., 2016*), putatively related to grid-cell population activity (*Doeller et al., 2010*), one might expect the pmEC to map spatial distances between object positions in our task. However, we did not observe an association of pattern similarity change in the entorhinal cortex with Euclidean or geodesic distances based on shortest paths between object positions. One potential explanation for the absence of evidence for a spatial distance signal in pmEC

might be the way in which we cued participants' memory during the picture viewing task. The presentation of isolated object images probed locations in their stored representation of the virtual city. Due to the periodic nature of grid-cell firing, different locations might not result in diverging patterns of grid-cell population activity. Hence, the design here was not optimized for the analysis of spatial representations in pmEC, if the object positions were encoded in grid-cell firing patterns as suggested by models of grid-cell function (*Fiete et al., 2008*; *Mathis et al., 2012*; *Bush et al., 2015*).

Our findings are in line with the role of the hippocampus in the retrieval of temporal information from memory (*Copara et al., 2014*; *DuBrow and Davachi, 2014*; *Kyle et al., 2015*; *Nielson et al., 2015*). Hippocampal pattern similarity has been shown to scale with temporal distances between events (*Deuker et al., 2016*; *Nielson et al., 2015*) and evidence for the reinstatement of temporally associated items from memory has been reported in the hippocampus (*DuBrow and Davachi, 2014*). Already at the stage of encoding, hippocampal and entorhinal activity have been related to later temporal memory (*DuBrow and Davachi, 2014*; *DuBrow and Davachi, 2016*; *Ezzyat and Davachi, 2014*; *Jenkins and Ranganath, 2010*; *Jenkins and Ranganath, 2016*; *Lositsky et al., 2016*; *Tubridy and Davachi, 2011*). For example, increased pattern similarity has been reported for items remembered to be close together compared to items remembered to be far apart in time, despite the same time having elapsed between these items (*Ezzyat and Davachi, 2014*). Similarly, changes in EC pattern similarity during the encoding of a narrative correlated with later duration estimates between events (*Lositsky et al., 2016*). Complementing these reports, our findings demonstrate that entorhinal activity patterns carry information about the temporal structure of memories at retrieval. Furthermore, the degree to which EC patterns reflected holistic representations of temporal relationships related to recall behavior characterized by the consecutive reproduction of objects encountered in temporal proximity; potentially through mental traversals of the route during memory recall. The central role of the hippocampus and entorhinal cortex in temporal memory (for review see *Davachi and DuBrow, 2015*; *Howard, 2018*; *Ranganath, 2019*; *Wang and Diana, 2017*) dovetails with the involvement of these regions in learning sequences and statistical regularities in general (*Barnett et al., 2014*; *Garvert et al., 2017*; *Hsieh et al., 2014*; *Kumaran and Maguire, 2006*; *Schapiro et al., 2012*; *Schapiro et al., 2016*; *Thavabalasingam et al., 2018*).

In this experiment, the paradigm was designed to disentangle temporal distances from Euclidean spatial distances between objects (*Deuker et al., 2016*), resulting also in a decorrelation of temporal distances and geodesic distances based on shortest paths between object positions. Since objects were encountered at regular intervals along the route, temporal distances quantified as elapsed time in seconds or, on an ordinal level of measurement, as the difference in sequence position were highly correlated measures of the sequence structure. To partially decorrelate elapsed time from ordinal temporal distances and distance traveled along the route, future studies could vary the speed of movement between different sections of the route. This might allow the investigation of the level of precision at which the hippocampal-entorhinal region stores temporal relations, in line with evidence for the integration of duration information in the representations of short sequences (*Thavabalasingam et al., 2018*; *Thavabalasingam et al., 2019*). Interestingly, a multi-scale ensemble of successor representations was recently suggested to estimate sequences of anticipated future states, including the order and distances between states (*Stachenfeld et al., 2017*; *Momennejad and Howard, 2018*); consistent with the sensitivity of neurons (*Sarel et al., 2017*; *Gauthier and Tank, 2018*; *Qasim et al., 2018*) and BOLD-responses (*Spiers and Maguire, 2007*; *Viard et al., 2011*; *Sherrill et al., 2013*; *Howard et al., 2014*; *Chrastil et al., 2015*) in the hippocampal-entorhinal region to distances and directions to navigational goals. Related to the effects of circumnavigation on travel time and Euclidean distance estimates (*Brunec et al., 2017*), experimental manipulations of route tortuosity could shed additional light on how, in the context of navigation, spatio-temporal event structures shape episodic memory.

In conclusion, our findings demonstrate that activity patterns in alEC, the human homologue region of the rodent lateral EC, carry information about the temporal structure of experienced events. The observed effects might be related to the reactivation of temporal contextual tags, in line with the recent report of temporal information available in rodent lateral EC population activity and models of episodic memory.

# Materials and methods

## Participants

26 participants (mean ± std. 24.88 ± 2.21 years of age, 42.3% female) were recruited via the university's online recruitment system and participated in the study. As described in the original publication using this dataset (*Deuker et al., 2016*), this sample size was based on a power-calculation (alpha-level of 0.001, power of 0.95, estimated effect size of d = 1.03 based on a prior study; *Milivojevic et al., 2015*) using G*Power (http://www.gpower.hhu.de/). Participants with prior knowledge of the virtual city (see *Deuker et al., 2016*) were recruited for the study. All procedures were approved by the local ethics committee (CMO Regio Arnhem Nijmegen, CMO2001/095, version 6.2) and all participants gave written informed consent prior to commencement of the study.

## Design

### Overview

The experiment began by a 10 min session during which participants freely navigated the virtual city (*Bellmund et al., 2018b*) on a computer screen to re-familiarize themselves with its layout. Afterwards participants were moved into the scanner and completed the first run of the picture viewing task during which they viewed pictures of everyday objects as described below (*Figure 1—figure supplement 1*). After this baseline scan, participants learned a fixed route through the virtual city along which they encountered the objects at predefined positions (*Figure 1* and *Figure 1—figure supplement 1*). The use of teleporters, which instantaneously moved participants to a different part of the city, enabled us to dissociate temporal from Euclidean and geodesic spatial distances between object positions (*Figure 1—figure supplement 2*). Subsequent to the spatio-temporal learning task, participants again underwent fMRI and completed the second run of the picture viewing task. Lastly, participants' memory was probed outside of the MRI scanner. Specifically, participants freely recalled the objects they encountered, estimated spatial and temporal distances between them on a subjective scale, and indicated their knowledge of the positions the objects in the virtual city on a top-down map (*Deuker et al., 2016*).

### Spatio-temporal learning task

Participants learned the positions of everyday objects along a trajectory through the virtual city Donderstown (*Bellmund et al., 2018b*). This urban environment, surrounded by a range of mountains, consists of a complex street network, parks and buildings. Participants with prior knowledge of the virtual city (see *Deuker et al., 2016*) were recruited for the study. After the baseline scan, participants navigated the fixed route through the city along which they encountered 16 wooden chests at specified positions (*Figure 1A*). During the initial six laps the route was marked by traffic cones. In later laps, participants had to rely on their memory to navigate the route, but guidance in the form of traffic cones was available upon button press for laps 7–11. Participants completed 14 laps of the route in total (mean ± standard deviation of duration 71.63 ± 13.75 min), which were separated by a black screen displayed for 15 s before commencement of the next lap from the start position.

Participants were instructed to open the chests they encountered along the route by walking into them. They were then shown the object contained in that chest for 2 s on a black screen. A given chest always contained the same object for a participant, with the assignment of objects to chests randomized across participants. Therefore, each object was associated with a spatial position defined by its location in the virtual city and a temporal position described by its occurrence along the progression of the route. Importantly, we dissociated temporal relationships between object pairs (measured by time elapsed between their encounter) from the Euclidean distance between their positions in the city through the use of teleporters. Specifically, at three locations along the route participants encountered teleporters, which immediately transported them to a different position in the city where the route continued (*Figure 1A*). This manipulation allows the otherwise impossible encounter of objects after only a short temporal delay, but with a large Euclidean distance between them in the virtual city (*Deuker et al., 2016*). Indeed, temporal distances across all comparisons of object pairs were not correlated with spatial relationships measured as Euclidean distances (*Figure 1—figure supplement 2A*).

An alternative way of capturing the spatial structure of the task is via geodesic distances. We quantified geodesic distances as the lengths of the shortest paths between object locations. Shortest paths were calculated using a Matlab implementation of the A* search algorithm (https://math-works.com/matlabcentral/fileexchange/56877). First, we calculated shortest paths that were allowed to cross all positions not obstructed by buildings or other obstacles (see *Figure 1—figure supplement 2D* for example paths). Second, because participants were instructed to only navigate on the streets during the learning task, we found shortest paths restricted to the city's street network (example paths are shown in *Figure 1—figure supplement 2E*). Neither form of geodesic distances between object positions was correlated with temporal distances (*Figure 1—figure supplement 2B, C*). Traveled-route distances were quantified as the median across laps of the distances participants traveled between the object positions when following the route.

## Picture viewing tasks

Before and after the spatio-temporal learning task, participants completed the picture viewing tasks while undergoing fMRI (*Deuker et al., 2016*). During these picture viewing tasks, the 16 objects from the learning task as well as an additional target object were presented. Participants were instructed to attend to the objects and to respond via button press when the target object was presented. Every object was shown 12 times in 12 blocks, with every object being shown once in every block. In each block, the order of objects was randomized. Blocks were separated by a 30 s break without object presentation. Objects were presented for 2.5 s on a black background in each trial and trials were separated by two or three TRs. These intertrial intervals occurred equally often and were randomly assigned to the object presentations. The presentation of object images was locked to the onset of the new fMRI volume. For each participant, we generated a trial order adhering to the above constraints and used the identical trial order for the picture viewing tasks before and after learning the spatio-temporal arrangement of objects along the route. Using the exact same temporal structure of object presentations in both runs rules out potential effects of temporal autocorrelation of the BOLD signal on the results, since such a spurious influence on the representational structure would be present in both tasks similarly and therefore cannot drive the pattern similarity *change* that we focused our analysis on (*Deuker et al., 2016*).

## MRI acquisition

All MRI data were collected using a 3T Siemens Skyra scanner (Siemens, Erlangen, Germany). Functional images during the picture viewing tasks were acquired with a 2D EPI sequence (voxel size 1.5 mm isotropic, TR = 2270 ms, TE = 24 ms, 40 slices, distance factor 13%, flip angle 85°, field of view (FOV) 210 × 210 × 68 mm). The FOV was oriented to fully cover the medial temporal lobes and if possible calcarine sulcus (*Deuker et al., 2016*). To improve the registration of the functional images with partial coverage of the brain, 10 volumes of the same functional sequence with an increased number of slices (120 slices, TR = 6804.1 ms) were acquired (see fMRI preprocessing). Additionally, gradient field maps were acquired (for 21 participants) with a gradient echo sequence (TR = 1020 ms; TE1 = 10 ms; TE2 = 12.46 ms; flip angle = 90°; volume resolution = 3.5 × 3.5×2 mm; FOV = 224 × 224 mm). Further, a structural image was acquired for each participant (voxel size = 0.8 × 0.8×0.8 mm, TR = 2300 ms; TE = 315 ms; flip angle = 8°; in-plane resolution = 256 × 256 mm; 224 slices).

## Quantification and statistical analysis

### Behavioral data

Results from in-depth analysis of the behavioral data obtained during the spatio-temporal learning task as well as the memory tests conducted after fMRI scanning are reported in detail in *Deuker et al. (2016)*. Here, we used data from the spatio-temporal learning task as predictions for multi-voxel pattern similarity (see below). Specifically, we defined the temporal structure of pairwise relationships between objects pairs as the median time elapsed between object encounters across the 14 laps of the route. These times differed between participants due to differences in navigation speed (*Deuker et al., 2016*). *Figure 1b* shows the temporal distance matrix averaged across participants for illustration. In our task, chests containing objects were spread evenly along the route and hence ordinal distances between objects provide a closely related measure of temporal structure

(mean ± standard deviation Pearson r = 0.993 ± 0.0014). For details of the analysis quantifying the relationship between entorhinal pattern similarity change and recall behavior see the corresponding section below.

## fMRI preprocessing

Preprocessing of fMRI data was carried out using FEAT (FMRI Expert Analysis Tool, version 6.00), part of FSL (FMRIB's Software Library, www.fmrib.ox.ac.uk/fsl, version 5.0.8), as described in *Deuker et al. (2016)*. Functional images were submitted to motion correction and high-pass filtering (cutoff 100 s). Images were not smoothed. When available, distortion correction using the fieldmaps was applied. Using FLIRT (*Jenkinson and Smith, 2001*; *Jenkinson et al., 2002*), the functional images acquired during the picture viewing tasks were registered to the preprocessed whole-brain mean functional images, which were in turn registered to the to the participant's structural scan. The linear registration from this high-resolution structural to standard MNI space (1 mm resolution) was then further refined using FNIRT nonlinear registration (*Anderson et al., 2007*). Representational similarity analysis of the functional images acquired during the picture viewing tasks was carried out in regions of interests co-registered to the space of the whole-brain functional images.

## ROI definition

Based on functional connectivity patterns, the anterior-lateral and posterior-medial portions of human EC were identified as human homologue regions of the rodent lateral and medial EC in two independent studies (*Navarro Schröder et al., 2015*; *Maass et al., 2015*). Here, we focused on temporal coding in the alEC, building upon a recent report of temporal signals in rodent lateral EC during navigation (*Tsao et al., 2018*). Therefore, we used masks from *Navarro Schröder et al. (2015)* to perform ROI-based representational similarity analysis on our data. For each ROI, the mask was co-registered from standard MNI space (1 mm) to each participant's functional space (number of voxels: alEC 126.7 ± 46.3; pmEC 69.0 ± 32.9). To improve anatomical precision for the EC masks, the subregion masks from *Navarro Schröder et al. (2015)* were each intersected with participant-specific EC masks obtained from their structural scan using the automated segmentation implemented in Freesurfer (version 5.3). ROI masks for the bilateral lateral occipital cortex were defined based on the Freesurfer segmentation and intersected with the combined brain masks from the two fMRI runs since this ROI was located at the edge of our field of view.

## Representational similarity analysis

As described in *Deuker et al., 2016*, we implemented representational similarity analysis (RSA, *Kriegeskorte et al., 2008a*; *Kriegeskorte et al., 2008b*) for the two picture viewing tasks individually and then analyzed changes in pattern similarity between the two picture viewing tasks, which were separated by the spatio-temporal learning phase. After preprocessing, analyses were conducted in Matlab (version 2017b, MathWorks). In a general linear model, we used the motion parameters obtained during preprocessing as predictors for the time series of each voxel in the respective ROI. Only the residuals of this GLM, that is the part of the data that could not be explained by head motion, were used for further analysis. Stimulus presentations during the picture viewing tasks were locked to the onset of fMRI volumes and the third volume after the onset of picture presentations, corresponding to the time 4.54 to 6.81 s after stimulus onset, was extracted for RSA.

For each ROI, we calculated Pearson correlation coefficients between all object presentations except for comparisons within the same of the 12 blocks of each picture viewing task. For each pairwise comparison, we averaged the resulting correlation coefficients across comparisons, yielding a 16 × 16 matrix reflecting the average representational similarity of objects for each picture viewing task (*Deuker et al., 2016*). These matrices were Fisher z-transformed. Since the picture viewing task was conducted before and after spatio-temporal learning, the two cross-correlation matrices reflected representational similarity with and without knowledge of the spatial and temporal relationships between objects, respectively. Thus, the difference between the two matrices corresponds to the change in pattern similarity due to learning. Specifically, we subtracted the pattern similarity matrix obtained prior to learning from the pattern similarity matrix obtained after learning, resulting in a matrix of pattern similarity change for each ROI from each participant. This change in similarity

of object representations was then compared to different predictions of how this effect of learning might be explained (*Figure 1B*).

To test the hypothesis that multi-voxel pattern similarity change reflects the temporal structure of the object encounters along the route, we correlated pattern similarity change with the temporal relationships between object pairs; defined by the participant-specific median time elapsed between object encounters while navigating the route. Likewise, we compared pattern similarity change to the Euclidean distances between object positions in the virtual city. We calculated Spearman correlation coefficients to quantify the fit between pattern similarity change and each prediction. We expected negative correlations as relative increases in pattern similarity are expected for objects separated by only a small distance compared to comparisons of objects separated by large distances (*Deuker et al., 2016*). We compared these correlation coefficients to a surrogate distribution obtained from shuffling pattern similarity change against the respective prediction. For each of 10000 shuffles, the Spearman correlation coefficient between the two variables was calculated, yielding a surrogate distribution of correlation coefficients (*Figure 1B*). We quantified the size of the original correlation coefficient in comparison to the surrogate distribution. Specifically, we assessed the proportion of larger or equal correlation coefficients in the surrogate distribution and converted the resulting p-value into a z-statistic using the inverse of the normal cumulative distribution function (*Deuker et al., 2016*; *Stelzer et al., 2013*; *Schlichting et al., 2015*). Thus, for each participant, we obtained a z-statistic reflecting the fit of the prediction to pattern similarity change in that ROI. For visualization (*Figure 2C*), we averaged correlation coefficients quantifying pattern similarity change in alEC separately for comparisons of objects encountered close together or far apart in time based on the median elapsed time between object pairs.

The z-statistics were tested on the group level using permutation-based procedures (10000 permutations) implemented in the Resampling Statistical Toolkit for Matlab (https://mathworks.com/matlabcentral/fileexchange/27960-resampling-statistical-toolkit). To test whether pattern similarity change in alEC reflected the temporal structure of object encounters, we tested the respective z-statistic against 0 using a permutation-based t-test and compared the resulting p-value against an alpha of 0.0125 (Bonferroni-corrected for four comparisons, *Figure 2*). Respecting within-subject dependencies, differences between the fit of temporal and spatial relationships between objects and pattern similarity change in the EC subregions were assessed using a permutation-based two-way repeated measures ANOVA with the factors EC subregion (alEC vs. pmEC) and relationship type (elapsed time vs. Euclidean distance). Planned post-hoc comparisons then included permutation-based t-tests of temporal against spatial mapping in alEC and temporal mapping between alEC and pmEC (Bonferroni-corrected alpha-level of 0.025).

## Accounting for adjacency effects

To rule out that only increased pattern similarity for object pairs encountered at adjacent temporal positions along the route drove the effect we excluded these comparisons from the analysis when testing whether pattern similarity change in alEC reflected temporal relationships. We tested the resulting z-values, reflective of holistic temporal maps independent of direct adjacency, against 0 (*Figure 2—figure supplement 1A*, one-sided permutation-based t-test). The z-values of this analysis were used for the correlation with recall behavior described below and shown in *Figure 2D*.

## Relationship between pattern similarity change and recall behavior

We assessed participants' tendency to reproduce objects encountered closely in time along the route at nearby positions during free recall. In this task, conducted after the post-learning picture viewing task, participants had two minutes to name as many of the objects encountered in the virtual city as possible and to speak the names in the order in which they came to mind into a microphone (*Deuker et al., 2016*). For each pair of recalled objects, we calculated the absolute positional difference in reproduction order and correlated these recall distances with elapsed time between object encounters of these pairs. This resulted in high Pearson correlation coefficients for participants with the tendency to recall objects at distant temporal positions along the route far apart and to retrieve objects encountered closely together in time along the route at nearby positions during memory retrieval. Such a temporally organized recall order would result for example from mentally traversing the route during the free recall task. The temporal organization of participants' free recall was

significantly correlated with the strength of the relationship between elapsed time and pattern similarity change after excluding comparisons of objects encountered at directly adjacent temporal positions (*Figure 2D*).

## Temporal intervals during the baseline scan

We interpret pattern similarity change between the picture viewing tasks as being induced by the learning task. To rule out effects of temporal intervals between objects experienced outside of the virtual city we correlated pattern similarity change in the alEC with temporal relationships during the pre-learning baseline scan. Specifically, we calculated the average temporal distance during the first picture viewing task for each pair of objects. Analogous to the time elapsed during the task, we correlated these temporal distances with pattern similarity change in the alEC. One participant was excluded from this analysis due to a z-value more than 1.5 times the interquartile range below the lower quartile. We tested whether pattern similarity change differed from zero and whether correlations with elapsed time during the task were more negative than correlations with temporal distance during the first picture viewing task (one-sided test) using permutation-based t-tests (*Figure 2—figure supplement 1B*).

## Timeline reconstruction

To reconstruct the timeline of events from alEC pattern similarity change we combined multidimensional scaling with Procrustes analysis (*Figure 2A*). We first rescaled the pattern similarity matrix to a range from 0 to 1 and then converted it to a distance matrix (distance = 1 − similarity). We averaged the distance matrices across participants and subjected the resulting matrix to classical multidimensional scaling. Since we were aiming to recover the timeline of events, we extracted coordinates underlying the averaged pattern distance matrix along one dimension. In a next step, we fitted the resulting coordinates to the times of object encounters along the route, which were also averaged across participants, using Procrustes analysis. This analysis finds the linear transformation, allowing scaling and reflections, that minimizes the sum of squared errors between the two sets of temporal coordinates. To assess whether the reconstruction of the temporal relationships between memories was above chance, we correlated the reconstructed temporal coordinates with the true temporal coordinates using Pearson correlation (*Figure 2B*). 95% confidence intervals were bootstrapped using the Robust Correlation Toolbox (*Pernet et al., 2012*). Additionally, we compared the goodness of fit of the Procrustes transform—the Procrustes distance, which measures the deviance between true and reconstructed coordinates—to a surrogate distribution. Specifically, we randomly shuffled the true temporal coordinates and then mapped the coordinates from multidimensional scaling onto these shuffled timelines. We computed the Procrustes distance for each of 10000 iterations. We quantified the proportion of random fits in the surrogate distribution better than the fit to the true timeline (i.e. smaller Procrustes distances) and expressed it as a p-value to demonstrate that our reconstruction exceeds chance level (*Figure 2C–D*).

## Signal-to-noise ratio

We quantified the temporal and spatial signal-to-noise ratio for each ROI. Temporal signal-to-noise was calculated for each voxel as the temporal mean divided by the temporal standard deviation for both runs of the picture viewing task separately. Values were averaged across the two runs and across voxels in the ROIs. Spatial signal-to-noise ratio was calculated for each volume as the mean signal divided by the standard deviation across voxels in the ROI. The resulting values were averaged across volumes of the time series and averaged across the two runs. Signal-to-noise ratios were compared between ROIs using permutation-based t-tests.

## Classification analysis

To examine whether object representations were stable between the pre- and the post-learning scan, we turned to pattern classification techniques and examined whether classifiers trained on the pre-learning scan exhibited systematic errors when tested on the post-learning data. Using the same time window as for the representational similarity analysis described above, we used data corresponding to the activation patterns evoked by individual object presentations during the picture viewing tasks from the LOC, alEC and pmEC. Data for each voxel within an ROI were z-scored

separately for the pre- and post-learning scan. For the pre-learning data of each ROI, we trained support vector machines on the binary classification of object identities in a one-versus-one coding design using the Matlab (version 2018b) function fitcecoc. Then, we tested the resulting classifiers on the independent data from the post-learning picture viewing task. We tested for stable object representations by comparing the percentages of correctly predicted object labels against chance with permutation-based one-sample t-tests (lag 0 in *Figure 2—figure supplement 4B*). Participant-specific chance levels were determined as average classifier accuracies when comparing classifier predictions to randomly permuted trial labels (1000 permutations).

In a second step, we examined classifier evidence as a function of the objects' positions along the route. If learned associations between objects lead to the reactivation of representations corresponding to objects from neighboring sequence positions, one might expect systematic classifier errors. We calculated classifier evidence for the three objects preceding and following a given object by shifting the true labels for each lag. At each lag, we excluded trials where shifted labels were invalid, that is not in the range of 1–16 for the 16 objects along the route, when calculating the percentage of hits. Chance levels were determined by randomly permuting the true labels for each lag. Classifier performance was tested against chance levels using permutation-based t-tests at each lag (*Figure 2—figure supplement 4B*). Note that classifier performance is below chance for some preceding and upcoming sequence positions due to high accuracy at lag 0.

## Acknowledgements

The authors would like to thank Raphael Kaplan for comments on a previous version of this manuscript. CFD's research is supported by the Max Planck Society; the European Research Council (ERC-CoG GEOCOG 724836); the Kavli Foundation, the Centre of Excellence scheme of the Research Council of Norway – Centre for Neural Computation, The Egil and Pauline Braathen and Fred Kavli Centre for Cortical Microcircuits, the National Infrastructure scheme of the Research Council of Norway – NORBRAIN; and the Netherlands Organisation for Scientific Research (NWO-Vidi 452-12-009; NWO-Gravitation 024-001-006; NWO-MaGW 406-14-114; NWO-MaGW 406-15-291). The funders had no role in study design, data collection and analysis, decision to publish or preparation of the manuscript.

## Additional information

### Funding

| Funder | Grant reference number | Author |
|---|---|---|
| Max-Planck-Gesellschaft | | Christian F Doeller |
| European Research Council | ERCCoG GEOCOG 724836 | Christian F Doeller |
| Nederlandse Organisatie voor Wetenschappelijk Onderzoek | NWO-Vidi 452-12- 009 | Christian F Doeller |
| Nederlandse Organisatie voor Wetenschappelijk Onderzoek | NWO-Gravitation 024-001-006 | Christian F Doeller |
| Nederlandse Organisatie voor Wetenschappelijk Onderzoek | NWO-MaGW 406-14-114 | Christian F Doeller |
| Nederlandse Organisatie voor Wetenschappelijk Onderzoek | NWO-MaGW 406-15-291 | Christian F Doeller |
| Kavli Foundation | | Christian F Doeller |
| Research Council of Norway | 223262 | Christian F Doeller |
| The Egil and Pauline Braathen and Fred Kavli Centre for Cortical Microcircuits | | Christian F Doeller |
| NORBRAIN – National Infrastructure scheme of the Research Council of Norway | | Christian F Doeller |

The funders had no role in study design, data collection and interpretation, or the decision to submit the work for publication.

### Author contributions
Jacob LS Bellmund, Conceptualization, Formal analysis, Investigation, Visualization, Writing—original draft, Writing—review and editing; Lorena Deuker, Conceptualization, Formal analysis, Investigation, Writing—review and editing; Christian F Doeller, Conceptualization, Supervision, Funding acquisition, Writing—review and editing

### Author ORCIDs
Jacob LS Bellmund (iD) https://orcid.org/0000-0002-2098-4487
Lorena Deuker (iD) http://orcid.org/0000-0002-4939-5862
Christian F Doeller (iD) https://orcid.org/0000-0003-4120-4600

### Ethics
Human subjects: All procedures were approved by the local ethics committee (CMO Regio Arnhem Nijmegen, CMO2001/095, version 6.2) and all participants gave written informed consent prior to commencement of the study.

### Decision letter and Author response
Decision letter https://doi.org/10.7554/eLife.45333.022
Author response https://doi.org/10.7554/eLife.45333.023

## Additional files

### Supplementary files
• Transparent reporting form
DOI: https://doi.org/10.7554/eLife.45333.019

### Data availability
Source data files have been provided for Figures 2, 3 and 4. The virtual city Donderstown is available at https://osf.io/78uph/.

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
