## [Decision Letter]

Thank you for submitting your article "Structuring time in human lateral entorhinal cortex" for consideration by *eLife*. Your article has been reviewed by three peer reviewers, including Ida Momennejad as the Reviewing Editor and Reviewer #2, and the evaluation has been overseen by Timothy Behrens as the Senior Editor. The following individuals involved in review of your submission have agreed to reveal their identity: Marc Howard (Reviewer #1); H Freyja Ólafsdóttir (Reviewer #3).

The reviewers have discussed the reviews with one another and the Reviewing Editor has drafted this decision to help you prepare a revised submission.

Summary:

Bellmund and colleagues describe interesting findings pointing to a role of the alEC in temporal coding. Specifically, the authors show objects encountered with close temporal proximity during a spatiotemporal task show similar pattern similarity using an fMRI RSA analysis. Importantly, the degree of temporal coding in the alEC correlated with behavioural performance. This paper examines changes in the representation of stimuli presented in a virtual world as a function of the time between and the (virtual) distance between their presentations. Previous work from the same group (using the same dataset as a matter of fact) and others has shown that learning causes representations of stimuli presented close together in time to become more similar to one another. This paper reexamines these data to specifically examine the role of subregions of the entorhinal cortex. This is inspired by a recent rodent study from the Moser group (Tsao et al., 2018) that has received a great deal of attention. The primary novel result is that there is an interaction of distance accounted for by time and space in the anterior-lateral entorhinal cortex (alEC) as compared to the posterior-medial entorhinal cortex (pmEC).

This paper is an advancement to previous work, an fMRI paradigm published in *eLife* by the same group. The authors ask a timely question about the representation of temporal structures in the human anterior lateral entorhinal cortex. The paper is very well written and the reasoning is clear. All reviewers agree that the manuscript presents a novel finding, which adds to the nascent body of research dedicated to studying the coding of space and time in the brain. Moreover, the work agrees with findings recently reported in rodents (Tsao et al., 2018). Overall, the reviewers find the manuscript timely and of broad interest and would like to see it published. Comments and revision requests are summarized below.

Essential Revisions:

Broadly: Is the finding clearly about time or something else (e.g. ordinal information, object reactivation, distance, etc.)?

Different reviewers had questions about the temporal nature of the findings and other interpretations. One of the reviewers notes that unlike the Tsao paper, this manuscript does not show direct evidence for a temporal representation in alEC. Here temporal (or related) information is incorporated into the response to a repeated stimulus perhaps due to recovery of temporal information – a jump back in time. The contrast to spatial information pmEC is quite meaningful and validates the relevance of a large body of rodent work for studies of human memory. These results are just as well predicted by models of human memory and localize the temporal vs. spatial aspect to the alEC vs. pmEC. Many models of human memory (including the temporal context model as well as a Laplace transform of time) hypothesize gradually-decaying activation of preceding information. This is also just what the Tsao Nature paper found. While one can choose to call this decaying trace of an object representation or a temporal code, the authors are advised to clarify that the temporal nature is not exclusively established early on, throughout the manuscript and by including the ordinal figure in the main results, and in the Discussion.

The comments are listed below. Unless the authors are inclined to conduct control experiments that verifies the 'temporal' nature of the findings, they are advised to revise the title as the present findings do not clearly delineate a temporal code. Detailed comments are listed below.

– The authors state their findings corroborate those of Tsao and colleagues (2018), which show temporal coding in the EC in rats. Specifically, Tsao et al. say they find EC codes for moment-to-moment changes in experience and explain the temporal signal may be less of a clocking signal and more of an experience timestamp.

Do the authors interpret their finding along the same lines? Moreover, the authors show their effect is replicated when they do an ordinal analysis. Thus, do the authors claim the alEC signals temporal order specifically rather than time or experience? It would be useful for the authors to elaborate on the interpretation of their effect especially in light of the interpretation given by Tsao et al. for the EC temporal coding.

– Regarding the analysis which included the pmEC (which is meant to be the human homologue to the MEC in rats), should one not expect to see a big change in pattern similarity for objects spatially close together? i.e. pmEC is meant to contain grid cells which code for space, rather than time (or at least more robustly than time). Could the authors comment on why they think they do not observe the reverse effect in pmEC compared to alEC?

– Another reason this is important regards the time scale of temporal distances in the Tsao experiment compared to the present study. Given the large time scales here, and the fact that the rodent electrophysiology data related to smaller time scales, it is possible that this result has to do with other measures/scales of distance. Please discuss.

– Do the authors believe that short/long time scales are coded by the same region? Do they think there's a gradient in alEC that allows computational of long and short temporal/ordinal distances?

– The authors consider "spatial distance" to only denote Euclidean distance. While the teleport condition clearly can distinguish between Euclidean spatial distance and other distances, it is not sufficient to infer temporal distance. For instance, path distance and geodesic distance are relevant here. This is a crucial point, since other sorts of distance such as path distance and communicability distance could be correlated with representational similarity results that the authors take to be uniquely indicative of time distance – depending on the sequence the participants observed.

This is a major concern: dissociating both Euclidean and non-Euclidean spatial distances from temporal distance requires control experiments and a bit of computational work.

Figure 2—figure supplement 1: This figure regarding the relevance of ordinal distances should be in the main text. In addition, please discuss the relationship between time and ordinal distance and compare them to other types of distance (path/geodesic, communicability, etc.). Specifically, if the correlation to other types of distance turned out to be significant that should be shown in the main text as well. Do other distance measure not previously discussed in the paper also capture the patterns in the alEC?

On a related note: in the Materials and methods, the analysis does not consider spatial distance per se, but merely *Euclidean* distance. While the analysis can dissociate Euclidean distance from ordinal/temporal distance, the authors thus cannot conclude that they have dissociated space and time. This section can be revised depending on the results of the analyses/simulations the authors would conduct involving other forms of distance mentioned above.

As a potential future experiment, if the speed of motion was controlled (e.g. participants were taken from A to B with fixed geodesic or ordinal or path or goal or communicability distances with controlled varying speeds) then it would be easier to decide whether the distance denotes time or some form of non-Euclidean spatial distance such as path distance (which will probably correlate with ordinal distance). Changing speed would make it clear whether path distance or geodesic or communicability is relevant here or time distance can explain the variance observed in the region of interest.

– While it is not necessary that the authors run the following analyses, here are some thoughts. It is possible that the authors have the required data to test the demands of the speed-controlled experiment. Specifically, since the experiments are self-paced, it is possible that the authors can compare pattern similarity in temporal distances separated by geodesic distance x speed variations. If the present experiment does not include sufficient number of trials per subject to make this analysis work, perhaps a super-group analysis may be possible (which the authors also are not required to perform). For instance, one could use SRM or other methods of pooling individual data into a group space and explore the nature of the distance x speed x time interaction.

Whether the authors run the previous analysis or not, this is an important point also because they report similarity to ordinal results, which will also be related to path distance, communicability distance, and other forms of distance measures that can be derived from a graph of the sequence or successor predictive representations (and can still be dissociated from Euclidean distance).

The authors could discuss their reasoning about why the potentially object trace, ordinal, or distance code finding is temporal in nature in the Discussion. They could discuss future experiments and analyses that can discern whether this is an exclusively temporal representation or it could be object race or path distance or other spatially relevant distance in their future directions (e.g. time can be discerned by varying speed across similar distances in a controlled fashion). However, making a strong claim about "time" in the title given the present findings and the role of ordinal information seems unwarranted.

---

## [Author Response]

Essential RevisionsBroadly: Is the finding clearly about time or something else (e.g. ordinal information, object reactivation, distance, etc.)?Different reviewers had questions about the temporal nature of the findings and other interpretations. One of the reviewers notes that unlike the Tsao paper, this manuscript does not show direct evidence for a temporal representation in alEC. Here temporal (or related) information is incorporated into the response to a repeated stimulus perhaps due to recovery of temporal information – a jump back in time. The contrast to spatial information pmEC is quite meaningful and validates the relevance of a large body of rodent work for studies of human memory. These results are just as well predicted by models of human memory and localize the temporal vs. spatial aspect to the alEC vs. pmEC. Many models of human memory (including the temporal context model as well as a Laplace transform of time) hypothesize gradually-decaying activation of preceding information. This is also just what the Tsao Nature paper found. While one can choose to call this decaying trace of an object representation or a temporal code, the authors are advised to clarify that the temporal nature is not exclusively established early on, throughout the manuscript and by including the ordinal figure in the main results, and in the Discussion.The comments are listed below. Unless the authors are inclined to conduct control experiments that verifies the 'temporal' nature of the findings, they are advised to revise the title as the present findings do not clearly delineate a temporal code. Detailed comments are listed below.

We would like to thank the reviewers for their positive evaluation of our manuscript and the constructive feedback. We appreciate the helpful comments and suggestions and have taken the opportunity to clarify our views on the points raised by the reviewers. In line with the reviewers’ suggestions we have refined the interpretation of our results in the revised manuscript. As pointed out correctly, we do not measure temporal information during the learning task. Rather, our analysis capitalizes on changes in multi-voxel activity patterns elicited by the object cues during the picture viewing tasks before and after learning. In our design, we present stimuli in random order during the picture viewing task and yet observe activity patterns that reflect the temporal structure experienced during learning. Indeed, this is consistent with the notion of a “jump back in time” elicited by the object representations. For example, our results could be explained by a reinstatement of temporal contextual tags or by a reactivation of objects encountered nearby in time. We agree that these explanations are in line with models of human memory. The findings by Tsao et al. (2018), who decode temporal information from neural signals during ongoing navigation behavior, but do not focus on memory per se, are of relevance by emphasizing the role of the lateral entorhinal cortex, specifically. This served as our motivation to analyze pattern similarity change in the entorhinal subdivisions separately.

We have followed the suggestion to incorporate the analysis using ordinal temporal distances in the main manuscript. Further, we more prominently discuss this finding and describe that ordinal distances also reflect the temporal structure of the event sequence, but that with the current design we cannot disentangle the precise scale of temporal representations. We have further adapted the title as suggested by the reviewers. The title of the manuscript is now “Mapping sequence structure in the human lateral entorhinal cortex”. As we describe below and throughout the manuscript, both elapsed time and ordinal temporal distances capture the temporal structure of the sequence participants experienced during the learning task. Please find our detailed responses to the individual comments below.

– The authors state their findings corroborate those of Tsao and colleagues (2018), which show temporal coding in the EC in rats. Specifically, Tsao et al. say they find EC codes for moment-to-moment changes in experience and explain the temporal signal may be less of a clocking signal and more of an experience timestamp.Do the authors interpret their finding along the same lines? Moreover, the authors show their effect is replicated when they do an ordinal analysis. Thus, do the authors claim the alEC signals temporal order specifically rather than time or experience? It would be useful for the authors to elaborate on the interpretation of their effect especially in light of the interpretation given by Tsao et al. for the EC temporal coding.

The reviewers ask the important question which role specifically the alEC might play for temporal coding. We appreciate the opportunity to offer our views on this issue. Tsao et al. (2018) emphasize that neural activity in the lateral subdivision of the entorhinal cortex in particular carries temporal information and suggest that this is due to the uniqueness of experience at every moment in time. In terms of the anatomical location of the effect, our data are in line with these findings in rodents as the anterior-lateral entorhinal cortex (alEC) is considered the human homologue region of the rodent LEC (Navarro Schröder et al., 2015; Maass et al., 2015). Further, they report cells with activity profiles that vary linearly over time. As alluded to above, no neural data was collected during the learning phase of our study. This precludes making strong interpretations about how specifically the effect we observe in the alEC relates to activity during the learning task. One explanation would be indeed that a slowly drifting signal, which might vary similarly on each lap due to the experience of navigating the same route (c.f. Tsao et al., 2018), provides timestamps for the object encounters during learning. These might in turn be reactivated in the alEC during the picture viewing task after learning, which could give rise to activity patterns reflecting the temporal distances of the task. This would be consistent with neural “jumps back in time” that have previously been observed in the human brain during image recognition tasks (Howard et al., 2012; Folkerts et al., 2018). Given the evidence for gradually drifting population activity in the human medial temporal lobe such an explanation seems more parsimonious to assume than an explicit clocking signal.

Our main analysis uses the median elapsed time between object encounters to show that alEC pattern similarity change correlates with the temporal structure of object relationships. We observe comparable results when quantifying the temporal structure of object relationships on an ordinal level of measurement using the difference in sequence position. We would like to stress that in our view such a representation of the serial order likely reflects a representation of the temporal structure of the encountered event sequence. One way how ordinal representations of the object sequence could arise is through object-to-object associations. For example, chaining models (e.g. Lewandowsky and Murdock, 1989; Jensen and Lisman, 2005) would predict that, during learning, an object is associated with preceding and successive objects in the sequence. Positions closer together in the sequence could result in stronger associations. During the picture viewing task after learning, object cues could result in the reactivation of associated objects. The strength of this reactivation might be more or less strong based on the strength of the association. This might give rise to activation patterns reflecting the serial order of events in a holistic fashion as observed in the alEC. Related to this question, we test in a new analysis in response to Comment 6 whether we can decode object representations in the entorhinal cortex and the lateral occipital cortex (LOC) using support vector machines. In brief, we do not observe stable object representations from the pre- to the post-scan picture viewing task in the entorhinal cortex.

While we could decode object identity from voxel patterns in the LOC, we did not observe evidence for the cortical reinstatement of temporally contiguous objects, which might be expected if the reactivation of object representations would underlie the effects we observed (see our response to Comment 6 for a detailed description of the analysis and results).

To dissociate whether temporal structure is represented on an ordinal as opposed to an interval or logarithmic level one would need an experimental design in which order and time/experience are at least partly decorrelated. Our experiment was designed to dissociate temporal and Euclidean spatial distances between objects and therefore the objects were spread fairly evenly along the route and participants’ movement speed was constant throughout the environment, which is why we cannot disentangle elapsed time from ordinal positions. In the revised manuscript, we have expanded our considerations of how a representation of the sequence structure might arise and how future studies might be able to dissociate ordinal temporal distances from time elapsed.

Please see below for the revised sections of the manuscript.

Introduction section

“This temporal information was suggested to arise from the integration of experience rather than an explicit clocking signal (Tsao et al., 2018).”

Discussion section

“While this interpretation is in line with data from rodent electrophysiology (Tsao et al., 2018) and the framework proposed by the temporal context model (Howard and Kahana, 2002; Howard et al., 2005) as well as evidence for neural contiguity effects in image recognition tasks (Howard et al., 2012; Folkerts et al., 2018), we cannot test the reinstatement of specific activity patterns from the learning phase directly since fMRI data were only collected during the picture viewing tasks in this study.”

“An alternative explanation for how the observed effects might arise is through associations between the objects. […] Hence, these results fail to provide evidence for the notion that the reactivation of object representations drove our effects.”

“In this experiment, the paradigm was designed to disentangle temporal distances from Euclidean spatial distances between objects (Deuker et al., 2016). […] This might allow the investigation of the level of precision at which the hippocampal-entorhinal region stores temporal relations, in line with evidence for the integration of duration information in the representations of short sequences (Thavabalasingam et al., 2018, 2019).”

– Regarding the analysis which included the pmEC (which is meant to be the human homologue to the MEC in rats), should one not expect to see a big change in pattern similarity for objects spatially close together? i.e. pmEC is meant to contain grid cells which code for space, rather than time (or at least more robustly than time). Could the authors comment on why they think they do not observe the reverse effect in pmEC compared to alEC?

The reviewers here ask the question why we did not observe reliable pattern similarity changes scaling with spatial distances in the posterior-medial subdivision of the entorhinal cortex in this task. This is a very interesting question given the wealth of literature describing spatial coding in the medial entorhinal cortex. Perhaps most prominently, grid cells have been discovered in the MEC (Hafting et al., Nature, 2005). Consistently, we have previously observed hexadirectional signals in the human pmEC (Bellmund et al., 2016), which are thought of as a proxy measure for activity in the entorhinal grid system in fMRI (Doeller et al., 2010). Models of grid-cell function suggest positions to be encoded by grid-cell population vectors in pmEC (e.g. Fiete et al., 2008; Mathis et al., 2012; Bush et al., 2015). Such spatial representations could be reactivated during the picture viewing task after learning in our study. However, cueing different positions might not result in BOLD activity patterns reflecting spatial distances. The analyses designed to detect grid-like entorhinal signals with fMRI are directional in nature, i.e. they contrast activity as a function of directions sampled in different trials (c.f. Doeller et al., 2010 for the univariate approach and Bellmund et al., 2016 for an adaptation of the analysis to multivoxel patterns). By presenting participants with isolated object images during picture viewing, we did not sample trajectories of different directions in this task and hence might not have been sensitive to spatial maps in pmEC if these were based on grid cell population codes.

We have made this notion explicit in the revised manuscript. The new section of the Discussion reads as follows:

“In line with hexadirectional signals in pmEC during imagination (Bellmund et al., 2016), putatively related to grid-cell population activity (Doeller et al., 2010), one might expect the pmEC to map spatial distances between object positions in our task. […] Hence, the design here was not optimized for the analysis of spatial representations in pmEC, if the object positions were encoded in grid-cell firing patterns as suggested by models of grid-cell function (Fiete et al., 2008; Mathis et al., 2012; Bush et al., 2015).”

– Another reason this is important regards the time scale of temporal distances in the Tsao experiment compared to the present study. Given the large time scales here, and the fact that the rodent electrophysiology data related to smaller time scales, it is possible that this result has to do with other measures/scales of distance. Please discuss.

The reviewers here ask the question how the time scale of our experiment compares to the experiment by Tsao et al. (2018), which decoded temporal information from LEC population signals. In our design, participants took around 264.6 ± 47.8s (mean ± standard deviation of median time per lap; see caption of Figure 1) to complete one lap of the route through the virtual city along which the 16 objects were encountered. In the rodent experiment, animals foraged for food in sessions consisting of a sequence of 12 trials with a trial length of 250s each. Tsao et al. show that they can not only decode trial identity, but also that they can decode shorter within-trial epochs. In their Figure 3F and Figure 3G, the authors show above chance decoding for epoch lengths of 20s, 10s and 1s. In our view, the trial length of 250s is comparable to the length of a lap of the route in our study. Further, epochs with a length of 10s and 20s constitute a similar temporal scale in comparison to our experiment, where objects were encountered on average every 16.6 ± 5.0s (mean ± standard deviation) on each lap. Therefore, we believe that the temporal scales can actually be regarded as comparable. Yet, of course, a key difference that remains is that Tsao et al. base their analyses on the period of ongoing activity, whereas our analyses focus on representations after learning. Nonetheless, the question of temporal scales in memory is intriguing as also discussed in response to the following comment. Further, in our view, this does not preclude other temporal scales in human memory, where temporal relations might also be represented on different levels or chunked in superordinate hierarchical structures such as different days or weeks.

We have made the match in the length of one lap in our design and the length of a trial in the paper by Tsao et al. explicit in the revised version of the manuscript.

Discussion section

“In our task, one lap of the route took approximately 4.5 minutes on average; comparable to the 250s-duration of a trial in Tsao et al. (2018).”

– Do the authors believe that short/long time scales are coded by the same region? Do they think there's a gradient in alEC that allows computational of long and short temporal/ordinal distances?

The reviewers here ask the question whether different time scales are represented by the same brain regions and whether there might be a gradient of temporal representation in the alEC. Relevant to the question how different time scales are encoded in the subregions of the hippocampal formation is the time cell literature. Time cells have been observed in the hippocampus (e.g. Pastalkova et al., 2008; MacDonald et al., 2011; Kraus et al., 2013; Mau et al., 2018) and medial entorhinal cortex (Kraus et al., Neuron, 2015) of the rodent brain during running in place. The length of the temporal intervals tiled by the sequential firing of time cells are typically in the range of seconds, constituting a neural code on a shorter temporal scale. Notably, the firing patterns of time cells with elevated firing at specific moments during the delay differ from firing patterns in the alEC where some cells’ activity varied linearly over time (Tsao et al., 2018).

Of note, studies investigating time cells typically employ highly-trained tasks including a repeated temporal delay, which constitutes a difference to temporal information derived from decaying traces of prior experience. Conceptually, we believe that slowly drifting population signals carrying temporal information arising from decaying traces of prior experience could more easily provide temporal context information for naturalistic episodic memory where temporal intervals are typically not repeated. Temporal information based on timestamps from a slowly varying signal might be a way to inherently encode temporal structure even without longer training procedures, a property desirable for episodic memory. Despite changes in the ensemble of time cells active over different sessions (Mau et al., 2018), it remains unclear whether time cells indeed also encode longer intervals. One recent study failed to detect time cells for a delay with a length of 60s (Sabariego et al., 2019).

A question that arises from this consideration is how time cells would behave in our task where one lap of the route is characterized by multiple intervals between the different events. One possibility might be that a long sequence of time cells could encode the temporal progression along the entire route. Alternatively, each interval between object encounters could be encoded by the same firing sequence since object encounters occurred at fairly regular temporal intervals along the route.

With respect to different granularities of representations there is evidence from studies in rodents for a granularity increase along the dorsoventral hippocampal axis in rodents. Place field size in the hippocampus and grid scale in the medial entorhinal cortex increase from more dorsal to ventral recording sites in the rodent brain (Kjelstrup et al., 2008; Barry et al., 2007; Stensola et al., 2012). Consistently, gradients in the scale of mnemonic networks (Collin et al., 2015) and fMRI voxel dynamics (Brunec, Bellana et al., 2018) have been documented along the hippocampal long axis in the human brain. To the best of our knowledge, there is no evidence for a gradient of representations in the lateral subdivision of the entorhinal cortex. Using fMRI, one possibility to investigate this question in future studies could be to scrutinize the particularly slow fluctuations of the BOLD signal in the EC (Lositsky et al., 2016) in more detail. Specifically, one could apply the analysis approach developed by Brunec et al. (Curr. Bio., 2018) and contrast time courses and temporal autocorrelation of voxels at different anatomical positions within the alEC. Given the small size of this region of interest, the increased spatial resolution of fMRI at 7T might be required. A gradient within the alEC could then be measured by greater similarity among voxel time courses and higher temporal autocorrelation in anterior compared to more posterior voxels (c.f. Brunec et al., 2018).

Discussion section

“Time cell ensembles change over minutes and days (Mau et al., 2018), but their firing has been investigated predominantly in the context of short delays in the range of seconds. […] How memories are represented at different temporal scales, which might be integrated in hierarchically nested sequences such as different days within a week, remains a question for future research.”

– The authors consider "spatial distance" to only denote Euclidean distance. While the teleport condition clearly can distinguish between Euclidean spatial distance and other distances, it is not sufficient to infer temporal distance. For instance, path distance and geodesic distance are relevant here. This is a crucial point, since other sorts of distance such as path distance and communicability distance could be correlated with representational similarity results that the authors take to be uniquely indicative of time distance – depending on the sequence the participants observed.This is a major concern: dissociating both Euclidean and non-Euclidean spatial distances from temporal distance requires control experiments and a bit of computational work.Figure 2—figure supplement 1: This figure regarding the relevance of ordinal distances should be in the main text. In addition, please discuss the relationship between time and ordinal distance and compare them to other types of distance (path/geodesic, communicability, etc.). Specifically, if the correlation to other types of distance turned out to be significant that should be shown in the main text as well. Do other distance measure not previously discussed in the paper also capture the patterns in the alEC?

*On a related note: in the Materials and methods, the analysis does not consider spatial distance per se, but merely* Euclidean *distance. While the analysis can dissociate Euclidean distance from ordinal/temporal distance, the authors thus cannot conclude that they have dissociated space and time. This section can be revised depending on the results of the analyses/simulations the authors would conduct involving other forms of distance mentioned above.*

As a potential future experiment, if the speed of motion was controlled (e.g. participants were taken from A to B with fixed geodesic or ordinal or path or goal or communicability distances with controlled varying speeds) then it would be easier to decide whether the distance denotes time or some form of non-Euclidean spatial distance such as path distance (which will probably correlate with ordinal distance). Changing speed would make it clear whether path distance or geodesic or communicability is relevant here or time distance can explain the variance observed in the region of interest.

The reviewers raise an important point and comment on different distance measures that might describe pattern similarity changes in the entorhinal cortex. Our paradigm was designed to dissociate Euclidean spatial distances and elapsed time between objects. We agree that there are other distance measures that can be used to quantify spatial relationships. In terms of the geodesic distance, we implemented two approaches of finding the shortest path between all pairs of object positions. First, we focused on all locations in the virtual city that were not obstructed by the collision volumes of virtual buildings, trees, or other objects distributed throughout the city and created a corresponding map of valid locations. Alternatively, one might consider only the streets as valid locations for navigation. Indeed, participants were instructed to stay on the streets during the learning phase and a prompt appearing on the screen reminded them to do so whenever they left the street network. Consequently, we created a second map in which only the streets of the virtual city were navigable positions. For either approach, we used a Matlab implementation of the A* search algorithm (https://mathworks.com/matlabcentral/fileexchange/56877) to find the shortest paths between all pairs of object positions. Examples of the resulting shortest paths are shown in Figure 1—figure supplement 2D and E. Geodesic distances were then quantified as the lengths of these shortest paths. Importantly, geodesic distances were not correlated with temporal distances measured as median elapsed time (Figure 1—figure supplement 2B and C; based on obstacles: mean and standard deviation of individual Pearson r=-0.061 ± 0.006, minimum p=0.414, correlation with averaged temporal distance: Pearson r=-0.061, p=0.505; based on streets: mean and standard deviation of individual Pearson r=-0.041 ± 0.006, minimum p=0.552, correlation with averaged temporal distance: Pearson r=-0.041 p=0.653). Entorhinal pattern similarity change was not significantly correlated with the geodesic distances based on obstacles in the virtual city (alEC: T(25)=0.82, p=0.436; pmEC: T(25)=0.73, p=0.479, Figure 2—figure supplement 2A) or the street network (alEC: T(25)=0.36, p=0.715; pmEC: T(25)=0.92, p=0.375, Figure 2—figure supplement 2B). Pattern similarity change in alEC was more strongly related to temporal than geodesic distances: akin to the main analyses, we conducted a 2x2 repeated measures ANOVA with the factors EC subregion and temporal vs. geodesic distances based on the obstacle map, revealing a significant interaction (main effect subregion: F(1,25)=5.18, p=0.031, main effect distance type:

F(1,25)=0.99, p=0.330, interaction: F(1,25)=6.96, p=0.014, post hoc comparison of temporal and geodesic distance in alEC: T(25)=-2.88, p=0.009). Likewise, we observed comparable results when using geodesic distances based on the street network (main effect subregion: F(1,25)=6.68, p=0.017, main effect distance type: F(1,25)=0.81, p=0.376, interaction: F(1,25)=4.30, p=0.048, post hoc comparison of temporal and geodesic distance in alEC: T(25)=-2.51, p=0.019). Taken together, these data highlight that pattern similarity change in alEC was not related to geodesic spatial distances between object positions.

The reviewers further point towards the path and communicability distance as a measure of spatial distances. In our analyses, the lengths of the paths between objects are almost perfectly correlated with the time elapsing between object encounters. This is due to the fact that travelled distances and elapsed time are identical unless participants take breaks from navigating. Since our measures of representational similarity are acquired before and after the navigation of the route, we are not sensitive to stops on individual laps of the route because we have to rely on measures describing central tendencies of participants’ behavior during the learning task. Because participants navigate the route repeatedly, consistent biases in stopping behavior across laps are unlikely. If one follows the notion that correlations between pattern similarity change in alEC and the temporal structure of the task arise not from a ticking clock, but from the association of objects with a slowly drifting contextual signal, e.g. through the decaying trace of prior experience, the travelled distances can be conceived of as an additional proxy measure of past experience that is closely related to elapsed time (mean and standard deviation of individual Pearson r=0.98 ± 0.005, all p<0.001; correlation with averaged temporal distance: Pearson r=0.98, p<0.001). However, there seems to be, to the best of our knowledge, little evidence in the literature that the human alEC or rodent LEC specifically would be involved in the mapping of distances travelled along a path in a spatial sense. Rather, keeping track of travelled distances is closely related to path integration for which grid cells found in the (posterior-) medial subdivision of the EC are thought to be of central importance in rodents and humans (Hafting et al., Nature, 2005; Gil et al., Nat. Neurosci. 2018; Chen et al., Curr. Bio., 2015; Stangl et al., Curr. Bio., 2018). In sum, the path distance along the route in our experiment and, more generally, travelled distances might be important contributors to the accumulation of experience in the context of spatial navigation. The reviewers discuss the experimental idea to dissociate the path distance from temporal distances (both elapsed time or ordinal distances) by controlling participants’ walking speed. As also discussed in response to Comments 2 and 7, we agree that variations of movement speed would be a suitable manipulation to disentangle the path distance from temporal relationships between positions in a future experiment.

In contrast, we do not think that the communicability distance offers a plausible measure for the object relationships in this task. While it would be possible to convert the street network and object positions into a graph structure, the communicability distance provides a suboptimal measure to quantify participants’ learning experience in this experiment in our view. This is because participants experienced the objects in a deterministic sequence by navigating along a fixed route that was designed to have little overlap. In fact, only a short section of one street was traversed twice on one lap of the route (see Figure 1, paths from chest 1 to 2 and chest 13 to 14) and no object was encountered on this stretch. BOLD-signals in the entorhinal cortex have been shown to be sensitive to communicability distances between nodes on a graph when stimulus sequences during learning reflect random walks along the underlying graph (Garvert et al., 2017). However, the ambiguity of the stimulus sequence constitutes a marked difference to the deterministic structure of the object sequence encountered along a route consisting almost exclusively of unique paths through the city. Therefore, we have not correlated communicability distances with pattern similarity change.

We have revised the manuscript according to the suggestions by the reviewers. We have incorporated the results of the analysis using ordinal temporal distances into the main manuscript. Further, we have included the analysis of geodesic distances in Figure 1—figure supplement 2 and Figure 2—figure supplement 2. We have carefully gone through the manuscript to specify where spatial distances are operationalized as Euclidean distances. Please see below for the revised sections of the manuscript.

See Figure 4, Figure 1—figure supplement 2B-E and Figure 2—figure supplement 2.

Introduction section

“We used representational similarity analysis of fMRI multi-voxel patterns in the entorhinal cortex to address the question how learning the structure of an event sequence shapes mnemonic representations in the alEC.”

Results section

“The temporal distance structure of the object sequence can be quantified as the elapsed time between object encounters or as ordinal differences between their sequence positions, which are closely related in our task. Spatial distances on the other hand can be captured by Euclidean or geodesic distances between positions. Importantly, we dissociated temporal from Euclidean and geodesic spatial object relationships through the use of teleporters along the route (Figure 1—figure supplement 2).”

“Pattern similarity change in alEC did not correlate significantly with Euclidean spatial distances (T(25)=0.81, p=0.420) and pattern similarity change in posterior-medial EC (pmEC) did not correlate with Euclidean (T(25)=0.58, p=0.583) or temporal (T(25)=1.73, p=0.089) distances.”

“Operationalizing the temporal structure in terms of the ordinal distances between object positions in the sequence yielded comparable results since our design did not disentangle time elapsed from ordinal positions as objects were encountered at regular intervals along the route. […]Furthermore, the interaction of the two-by-two repeated measures ANOVA with the factors entorhinal subregion and distance type remained significant when using geodesic spatial distances based on shortest paths using all non-obstructed positions (interaction: F(1,25)=6.96, p=0.014; main effect of EC subregion: F(1,25)=5.18, p=0.031; main effect of distance type: F(1,25)=0.99, p=0.330) or the street network only (interaction: F(1,25)=4.30, p=0.048; main effect of EC subregion: F(1,25)=6.68, p=0.017; main effect of distance type: F(1,25)=0.81, p=0.376).”

Discussion

“In our task, relevant factors contributing to a similar experience of the route on each lap are not only the encounters of objects in a specific order at their respective positions, but also recognizing and passing salient landmarks as well as travelled distance and navigational demands in general.”

Materials and methods

“The use of teleporters, which instantaneously moved participants to a different part of the city, enabled us to dissociate temporal from Euclidean and geodesic spatial distances between object positions (Figure 1—figure supplement 2).”

“Indeed, temporal distances across all comparisons of object pairs were not correlated with spatial relationships measured as Euclidean distances (Figure 1—figure supplement 2A).”

“An alternative way of capturing the spatial structure of the task is via geodesic distances. We quantified geodesic distances as the lengths of the shortest paths between object locations. Shortest paths were calculated using a Matlab implementation of the A* search algorithm (https://mathworks.com/matlabcentral/fileexchange/56877). First, we calculated shortest paths that were allowed to cross all positions not obstructed by buildings or other obstacles (see Figure 1—figure supplement 2D for example paths). Second, because participants were instructed to only navigate on the streets during the learning task, we found shortest paths restricted to the city’s street network (example paths are shown in Figure 1—figure supplement 2E). Neither form of geodesic distances between object positions was correlated with temporal distances (Figure 1—figure supplement 2BC).”

– While it is not necessary that the authors run the following analyses, here are some thoughts. It is possible that the authors have the required data to test the demands of the speed-controlled experiment. Specifically, since the experiments are self-paced, it is possible that the authors can compare pattern similarity in temporal distances separated by geodesic distance x speed variations. If the present experiment does not include sufficient number of trials per subject to make this analysis work, perhaps a super-group analysis may be possible (which the authors also are not required to perform). For instance, one could use SRM or other methods of pooling individual data into a group space and explore the nature of the distance x speed x time interaction.Whether the authors run the previous analysis or not, this is an important point also because they report similarity to ordinal results, which will also be related to path distance, communicability distance, and other forms of distance measures that can be derived from a graph of the sequence or successor predictive representations (and can still be dissociated from Euclidean distance).The authors could discuss their reasoning about why the potentially object trace, ordinal, or distance code finding is temporal in nature in the Discussion. They could discuss future experiments and analyses that can discern whether this is an exclusively temporal representation or it could be object race or path distance or other spatially relevant distance in their future directions (e.g. time can be discerned by varying speed across similar distances in a controlled fashion). However, making a strong claim about "time" in the title given the present findings and the role of ordinal information seems unwarranted.

The reviewers here offer interesting suggestions for additional analyses to attempt to dissociate elapsed time from spatial distances. Unfortunately, the data we have from this experiment does not allow us to analyze the relationship of pattern similarity change and temporal distances for different walking speeds. The learning task was indeed self-paced, so there are variations in navigation efficiency and potentially walking speeds across laps. However, the representational change is only assessed after the learning task. Hence, we do not have a lap-by-lap measure of object representations that would allow more fine-grained analysis. Rather, we have to select one variable from the learning task and relate representational change to it. Performing an analysis where temporal distances are assessed as a function of speed would require data in which speed variations are consistent across laps by a participant. This would require the explicit experimental manipulation of walking speeds between different objects in a new experiment. We describe this idea for a future study in the Discussion.

The reviewers suggest to additionally discuss the potential role of object traces or predictive representations as possible explanations for the results. Since objects were presented in random order during the picture viewing tasks, memory-based reactivations of objects preceding or following a cued object would be required to explain the observed effect. In new analyses, we aimed to test predictions that can be derived from the idea that the observed effects reflect the, potentially predictive, reactivation of neighboring object representations. If object cues during the post-learning picture viewing task were to reactivate neighboring objects from a learned sequence, we should be able to detect such object representations. To test this idea, we turned towards a classification analysis. Using the Matlab function fitcecoc, we trained support vector machines in a one-vs.-one coding design to distinguish activity patterns evoked by the different object cues in different regions of interest. In addition to the entorhinal cortex, these included the lateral occipital cortex (LOC) – known to be involved in the visual processing of objects (for review see Grill-Spector et al., 2001) – to test for cortical reinstatement of object representations. We trained the classifiers on the data from the pre-learning scan to test for similar representations during the post-learning scan.

Using this procedure, we were able to classify object identities significantly above chance levels determined through random permutations of trial labels in the LOC (T(25)=7.54, p<0.001, Figure 2—figure supplement 4). However, decoding accuracies were at chance level for both entorhinal ROIs (alEC: T(25)=-0.08, p=0.941; pmEC: T(25)=0.53, p=0.621). If object images during the post-learning scan serve as cues for the reactivation of neighboring objects, objects n-1 and n+1 might be activated when viewing object n. This might result in systematic errors of the classifier. Hence, we analyzed the classifier evidence as a function of sequence position lag to test for above-chance classifier confusion using one-sided t-tests. Focusing on lags ± 3, we did not observe any above-chance classifier evidence for preceding (alEC: most extreme T(25)=1.13; minimum p=0.270; pmEC: most extreme T(25)=1.00; min. p=0.332) or upcoming (alEC: most extreme T(25)=-2.07; min. p=0.055; pmEC: most extreme T(25)=-0.83; min. p=0.414) objects in the entorhinal cortex. Based on evidence for the reinstatement of cortical activity patterns during retrieval (Nyberg et al., 2000; Wheeler et al., 2000; Polyn et al., 2005), which is modulated by hippocampalentorhinal activity (Bosch et al., 2014), we also performed this analysis in the LOC, but again failed to observe above-chance evidence for preceding or following objects. Rather classifier evidence was below chance levels, potentially due to the high accuracy at no lag (preceding objects: most extreme T(25)=-2.51; min. p=0.018; successive objects: most extreme T(25)=-4.09; min. p<0.001). Taken together, the lack of stable object representations in the entorhinal cortex makes it unlikely that the effects we observed go back to a decaying object trace or the reactivation of objects preceding or following a presented object along the route.

As noted above already, we have followed the suggestion to revise the title of the manuscript, which now reads “Mapping sequence structure in the human lateral entorhinal cortex”. Throughout the manuscript, we describe that the temporal structure of the object sequence can be quantified at different levels of measurement. Again, we would like to emphasize that, in our view, ordinal-level representations of the sequence also reflect the temporal structure of the object sequence.

In the revised manuscript, we have included the classification analyses and discuss potential future experiments dissociating elapsed time from ordinal temporal distances and spatial distances such as the path distance. The new figures and revised sections of the manuscript are:

Title: Mapping sequence structure in the human lateral entorhinal cortex

Figure 2—figure supplement 4

Results section

“Does the presentation of object images during the post-learning picture viewing task elicit reactivations of similar representations from the pre-scan? […] We also did not observe abovechance classifier evidence for nearby objects in the LOC, but rather classifier evidence was below chance levels for some lags, potentially due to high classification accuracies at no lag (preceding objects: most extreme T(25)=-2.51; min. p=0.018; successive objects: most extreme T(25)=-4.09; min. p<0.001).”

“An alternative explanation for how the observed effects might arise is through associations between the objects. […] Hence, these results fail to provide evidence for the notion that the reactivation of object representations drove our effects.”

Materials and methods

“ROI masks for the bilateral lateral occipital cortex were defined based on the Freesurfer segmentation and intersected with the combined brain masks from the two fMRI runs since this ROI was located at the edge of our field of view.”

“Classification analysis

To examine whether object representations were stable between the pre- and the postlearning scan, we turned to pattern classification techniques and examined whether classifiers trained on the pre-learning scan exhibited systematic errors when tested on the post-learning data. […] Note that classifier performance is below chance for some preceding and upcoming sequence positions due to high accuracy at lag 0.